# Persistently increased post-stress activity of paraventricular thalamic neurons is essential for the emergence of stress-induced alterations in behaviour

Anna Jász[1,2☯], László Biró[1☯]*, Zsolt Buday[1,2], Bálint Király[3,4], Orsolya Szalárdy[5], Krisztina Horváth[6], Gergely Komlósi[1], Róbert Bódizs[5], Krisztina J. Kovács[6], Marco A. Diana[7], Balázs Hangya[3], László Acsády[1]*

1 Lendület Laboratory of Thalamus Research, HUN-REN Institute of Experimental Medicine, Budapest, Hungary, 2 Neurosciences PhD School, Semmelweis University, Budapest, Hungary, 3 Lendület Laboratory of Systems Neuroscience, HUN-REN Institute of Experimental Medicine, Budapest, Hungary, 4 Department of Biological Physics, Institute of Physics, Eötvös Loránd University, Budapest, Hungary, 5 Psychophysiology and Chronobiology Research Group, Institute of Behavioral Sciences, Semmelweis University, Budapest, Hungary, 6 Laboratory of Molecular Neuroendocrinology, Institute of Experimental Medicine, Budapest, Hungary, 7 Université Paris Cité, CNRS, Saint-Pères Paris Institute for the Neurosciences, Paris, France

☯ These authors contributed equally to this work.
* biro.laszlo@koki.hu (LB); acsady@koki.hu (LA)

**Data Availability Statement:** All relevant data are within the paper and its Supporting Information

## Abstract

A single exposure to a stressful event can result in enduring changes in behaviour. Long-term modifications in neuronal networks induced by stress are well explored but the initial steps leading to these alterations remain incompletely understood. In this study, we found that acute stress exposure triggers an immediate increase in the firing activity of calretinin-positive neurons in the paraventricular thalamic nucleus (PVT/CR+) that persists for several days in mice. This increase in activity had a causal role in stress-induced changes in spontaneous behaviour. Attenuating PVT/CR+ neuronal activity for only 1 h after the stress event rescued both the protracted increase in PVT/CR+ firing rate and the stress-induced behavioural alterations. Activation of the key forebrain targets (basolateral amygdala, prelimbic cortex, and nucleus accumbens) that mediate defensive behaviour has also been reduced by this post-stress inhibition. Reduction of PVT/CR+ cell activity 5 days later remained still effective in ameliorating stress-induced changes in spontaneous behaviour. The results demonstrate a critical role of the prolonged, post-stress changes in firing activity of PVT/CR+ neurons in shaping the behavioural changes associated with stress. Our data proposes a therapeutic window for intervention in acute stress-related disorders, offering potential avenues for targeted treatment strategies.

## Introduction

A single stress event can alter behaviour for prolonged periods (months or years) and can lead to changes in the structure and function of various brain regions long after the stress event

files. Source codes used for analyzing the data are available at https://zenodo.org/records/14222815.

**Funding:** LA was supported by the European Research Council (ERC Advanced Grant, FRONTHAL, grant no. 742595; see details at https://erc.europa.eu/funding/advanced-grants) and the European Union within the framework of the Artificial Intelligence National Laboratory (RRF-2.3.1-21-2022-00004; more information at https://ai-hu.eu). BH was supported by the European Research Council (ERC Starting Grant, CholAminCo, grant no. 715043; details available at https://erc.europa.eu/funding/starting-grants) and the National Research, Development and Innovation Office of Hungary (NKFIH, grant no. K135561; information can be found at https://nkfih.gov.hu). LA also received funding from the "Lendület" Program of the Hungarian Academy of Sciences (grant no. LP2023-2/2023; further details at https://mta.hu/lendulet). RB was supported by the Ministry of Innovation and Technology of Hungary through the Higher Education Institutional Excellence Program (grant no. TKP2021-EGA-25; information available at https://nkfih.gov.hu). MAD received support from the French National Research Agency (ANR, grant nos. ANR-21-CE37-0025-02 and ANR-21-CE16-0012-02; information can be found at https://anr.fr). The funders had no role in study design, data collection and analysis, decision to publish, or preparation of the manuscript.

**Competing interests:** The authors have declared that no competing interests exist.

**Abbreviations:** BBS, bicarbonate-buffered solution; BNST, bed nucleus of the stria terminalis; CA, correlated activity; CORT, corticosterone; CR+, calretinin-positive; HFA, high-frequency activity; HPA, hypothalamic–pituitary–adrenal axis; IPSC, inhibitory postsynaptic current; mPFC, medial prefrontal cortex; NAc, nucleus accumbens; NDS, normal donkey serum; NOE, no odour exposure; NREM, non-rapid eye movement; PBS, phosphate buffer saline; POSE, predatory odour stress exposure; PVH, paraventricular hypothalamic nucleus; PVT, paraventricular thalamic nucleus; RIA, radioimmunoassay; ROI, region of interest; SALT, stimulus-associated spike latency test; SD, standard deviation; sIPSC, spontaneous inhibitory synaptic current; SMI, state modulation index.

[1,2]. However, the initial steps leading to these chronic changes have been very little studied [1,3]. If stress exerts persistent effects on behaviour one may assume that it is supported by a long-term alteration in firing activity. However, the mean firing rate is one of the most stable features of the neurons [4–8]. Neurons can significantly alter their activity in specific behavioural contexts (e.g., during a working memory task or sustained attention) [9–11], but these epochs usually last no longer than seconds, or in exceptional cases for minutes [12]. At the end of this so-called "persistent activity" neurons return to their basal firing rate. Altered, spontaneous firing rates for extended periods (i.e., days) have not been described so far, so it is presently unclear whether prolonged alteration of firing activity can underlie the persistent change in behaviour after stress.

In this study, we examined long-term changes in stress-induced neuronal activity at a key node of the stress circuit, the paraventricular thalamic nucleus (PVT), and studied its role in the alteration of spontaneous behaviour after stress. The PVT has long been acknowledged for its robust activation in response to various stressors, as evidenced by the immediate early genes expression studies [13–16]. The link between PVT and stress has predominantly been explored in the context of repeated, chronic stress models and conditional fear paradigms [14,17–20]. The outcomes from chronic stress models have yielded conflicting perspectives. While certain studies demonstrate that PVT lesion blocks the reduced behavioural and hormonal responses to repeated stress [17,21–23], others show that inhibiting PVT following intense restraint stress (lasting 2 h) can normalise behaviour in tasks sensitive to stress [18]. In paradigms involving fear memory retrieval, PVT was found to display a delayed role (24 h later) in mediating responses to conditioned stimuli, via top-down influence on PVT from cortical regions [16,24].

PVT is known to be sensitive to arousal and to salient events [13,25,26]. As a result of these features, interruption of PVT activity during the exposure to a stressor diminishes the magnitude of stress response and, as a consequence, the post-stress behavioural alterations [20]. In order to avoid this effect in the present study, we perturbed PVT activity only after the exposure to the stress, ensuring that the stress experience is unaltered across the experimental groups.

The largest neurochemically labelled cell population of the PVT consists of the calretinin-positive (CR+) neurons [13,27–30]. CR regulates intracellular calcium levels in neurons, modulating excitability and firing rates, which can support sustained neural activity and stability during prolonged periods of heightened activity [31,32]. PVT/CR+ neurons have been shown to receive selective inputs from several hypothalamic and brainstem centres both in rodents and humans [13,17,33,34]. The vast majority of PVT neurons that innervate the amygdala (Amy) medial prefrontal cortex (mPFC) or nucleus accumbens (NAc) express CR (over 94% of the projecting cells). These PVT/CR+ cells can exert fast, powerful postsynaptic effect on their forebrain targets [13]. PVT neurons are known to be activated by a wide variety of stressors [17] but within the PVT almost all of cells (over 97%) that respond to various levels of stress by c-Fos expression contain CR [13]. Finally, 93.5% of the PVT/CR+ but not the PVT/CR- neurons display a predictive increase in firing rate before behavioural arousal and have diurnal alterations in c-Fos levels [13]. These morphological and physiological data clearly suggest that PVT/CR+ cells represent a functionally relevant cell population within the PVT and that they can form a critical bottleneck in the brainstem-forebrain transfer of stress and arousal activity.

In this study, we asked whether PVT/CR+ neurons can undergo a prolonged alteration of firing activity once the stress exposure is over and whether this change can be causally linked to stress induced modifications of behaviour. We found that the activity of PVT/CR+ neurons display a persistent increase following a stress event for several days. Post-stress manipulation of PVT/CR+ neuronal activity demonstrated that the prolonged alteration of PVT/CR+ neurons is causally involved in the manifestation of post-stress behaviour.

## Results

### Short-term effects of predator odour stress exposure (POSE)

In this study, we used a well-established, behaviourally relevant model of stress, predatory odour stress exposure (POSE) [35–38]. We analysed the changes of spontaneous behaviour in the home cage of the animals and compared the pre- and post-stress periods in order to avoid interference of different stressors by exposing the animals to another stressful situation (e.g., elevated plus maze). We utilised optogenetic perturbation of PVT/CR+ neuronal activity timed after the stress event. Our manipulations and recordings were centred on middle portion of PVT (AP, BR: −0.8 mm to 1.8 mm) and did not include the rostral 500 μm sector of the nucleus or its most caudal end (see Methods section for).

After recording the pre-stress, behaviour in the home cage (PRE 1–5 days) we exposed mice to 2-methyl-2-thiazoline (2MT) for 10 min [39,40] in a novel cage and compared home cage activity between pre- and post-stress days. In order to test the role of post-stress PVT/CR+ neuronal activity in stress-induced alteration of behaviour, we perturbed PVT/CR+ firing after POSE (1 h) when the animals were returned to their home cage. We used intermittent activation (2 s ON, 13 s OFF) of the inhibitory opsin SwiChR in CR-Cre mice that was previously proved to be effective [13] (Figs 1A and S1). Since we applied optogenetic inhibition after, not during, POSE we avoided interference with perceiving the stress exposure. Tetrode recordings were used to demonstrate the effect of SwiChR activation. PVT/CR+ cells decreased their firing rate during the laser ON periods (S1 Fig), as expected. As a control for the SwiChR group, we used AAV-DIO-EYFP injected stressed animals (EYFP or non-inhibited group, Fig 1A and 1B). As a control for the stress situation, we used a no odour exposure (NOE) group that was exposed to the novel cage without 2MT and a home cage groups which remained in the home cage (see Methods) as well.

Both the EYFP and the SwiChR groups displayed robust defensive behaviours [41] (escape jumps and freezing) during the 10-min exposure to 2MT (Figs 1B–1E and S2A and S1 Movie) since the SwiChR group was not inhibited during the POSE. Freezing accounted for 95% of the total time spent in defensive behaviours. An increase in defensive behaviours was not observed in the NOE group (Fig 1F and S2 Movie) showing that the novel environment per se did not contribute to the defensive behaviours. Following the return of the animals to their home cage, on the day of the stress (POST0 day), EYFP mice displayed significantly more high-frequency respiration (hyperventilation, Fig 1G and S3 Movie), and a trend to elevated sleep onset latency (Fig 1H) compared to the SwiChR group (Fig 1H and S4 Movie) which underwent 1 h long photoinhibition after POSE. This indicates that inhibition of PVT/CR+ neurons after the stress event can ameliorate the immediate impact of stress on behaviour. During the optogenetic inhibition, the SwiChR animals did not display overt behavioural alterations, they showed normal locomotor activity (S4 Movie). However, agreeing with a previous report [13], they moved less compared to the identical period of their last pre-stress day (PRE5) (Fig 1I).

Next, we tested the impact of POSE and post-stress photoinhibition on c-Fos activation and hormonal changes. One hour after POSE PVT/CR+ neurons, the EYFP animals displayed significantly increased c-Fos expression (Fig 1J and 1K) relative to home cage and NOE controls. In the EYFP mice, c-Fos expression also increased in the paraventricular hypothalamic nucleus (PVH) (Fig 1L) together with significantly elevated blood corticosterone (CORT) levels (Fig 1M) as in earlier studies [42,43]. Photoinhibition of PVT/CR+ neurons for 1 h after stress abolished the increase in c-Fos expression of PVT/CR+ cells (Fig 1K). A decrease in c-Fos expression was most prominent under the fibre optic targeting PVT (S2B and S2C Fig). Activation of SwiChR in PVT/CR+ cells, however, did not result in decreased c-Fos activity in PVH (Fig 1L)

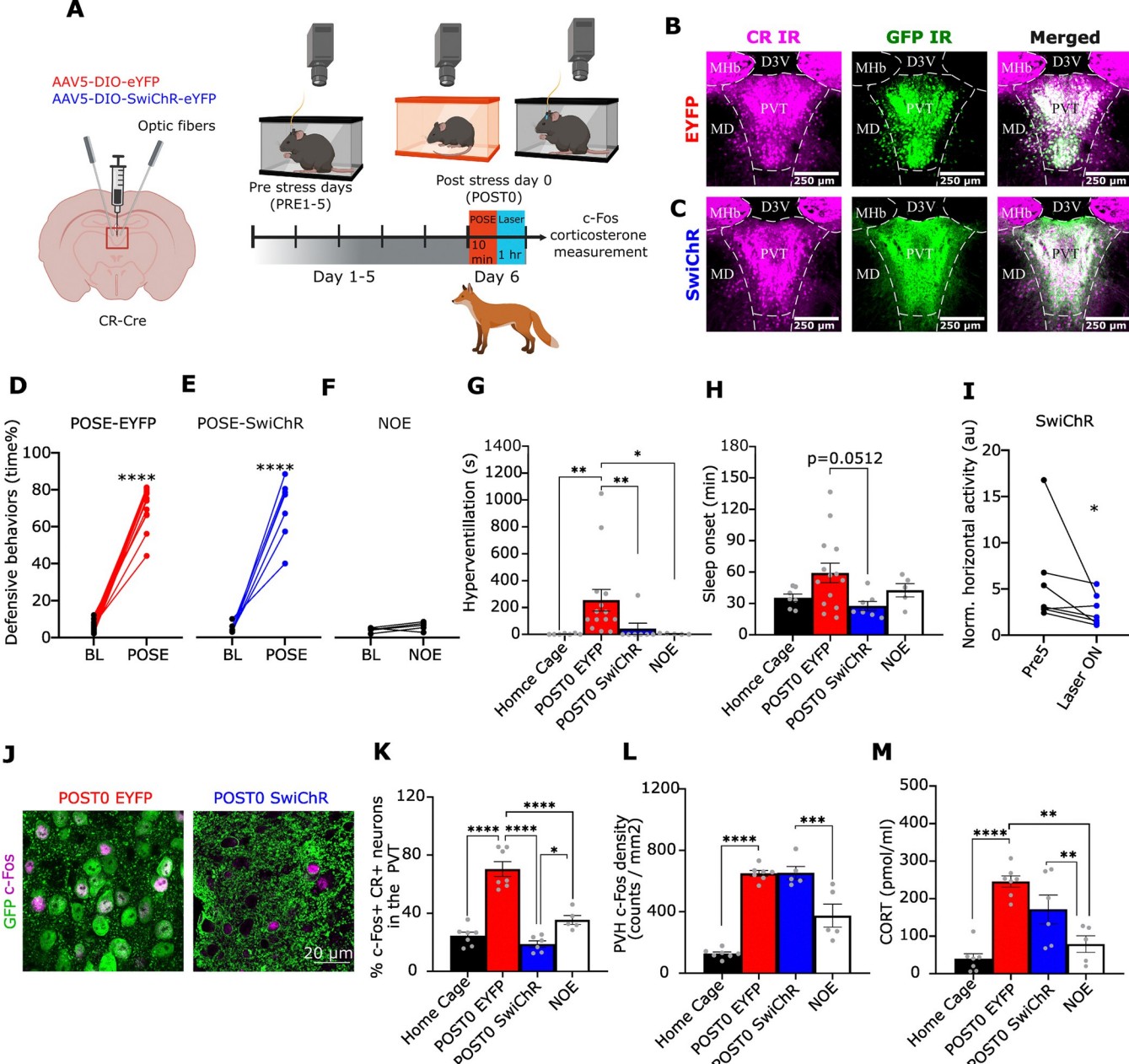

**Fig 1. Short term effects of inhibiting PVT/CR+ neurons after the predatory odour stress exposure.** (A) Scheme of the experiment. AAVs were injected into PVT and behaviour was assessed 4 weeks later. Following a 5 day baseline period to assess home cage behaviours (PRE5), mice were submitted to POSE. Immediately after the POSE both SwiChR and EYPF groups received a 1 h long, photoinhibition of PVT/CR+ neurons in the home cage. Created with [BioRender.com](BioRender.com). (B, C) Representative confocal images showing EYFP and SwiChR expression in CR-positive (CR+) PVT neurons. Magenta represents CR immunostaining, while green represents GFP immunostaining, respectively. MD, mediodorsal thalamic nucleus; Hab, habenula; D3V, third ventricle. Scale bars, 250 μm. (D, E) Quantification of time spent with defensive behaviour during predator odour (2MT) stress exposure (POSE) in EYFP (D, t[13] = 21.05) and SwiChR (E, t[6] = 9.35) mice compared to a 2 min baseline (BL) period. (F) Same for the NOE (without 2MT presentation) group (t[4] = 1.822). (G) Bar graph showing the duration of hyperventilation in the Home Cage control group ($n = 7$), EYFP ($n = 7$), SwiChR ($n = 6$), and NOE control ($n = 5$) in their home cage after POSE (K-W = 21.58, $p < 0.0001$). The Home Cage control group was not exposed to the novel environment. (H) Bar graph showing the duration of sleep onset in the Home Cage control group ($n = 7$), EYFP ($n = 7$), SwiChR ($n = 6$), and NOE ($n = 5$) in their home cage after POSE (F(3,29) = 2.961, $p < 0.0486$). (I) Averaged, normalised horizontal activity during PVT/CR+ photoinhibition in the SwiChR animals (first hour of POST0 day) compared to the same period of the PRE 5 day (W = −26). (J) Confocal images showing c-Fos expression in PVT/CR+ neurons in EYFP and SwiChR mice 1 h after POSE. Note that EYFP is a cytosolic labelling, whereas SwiChR is expressed in membranes. (K, L) Quantification of c-Fos expression in the PVT/CR+ cells (K, F(3,21) = 44.75, $p < 0.0001$) and in the paraventricular nucleus of the hypothalamus (PVH) (L) 1 h after POSE in the 4 experimental groups; Home Cage control ($n = 7$); EYFP ($n = 7$); SwiChR ($n = 6$); NOE control (F(3,20) = 50.00, $p < 0.0001$). (M) CORT level in the blood of the 4 experimental groups 1 h after POSE (F(3,21) =

16.96, $p < 0.0001$). Underlying data can be found in S1 Data. See S10 Data for the full results of the statistical tests. Data are shown as mean ± SEM. *$p < 0.05$, **$p < 0.01$, ***$p < 0.01$, ****$p < 0.001$. CORT, corticosterone; NOE, no odour exposure; POSE, predatory odour stress exposure; PVH, paraventricular hypothalamic nucleus; PVT, paraventricular thalamic nucleus.

or decreased blood corticosterone levels (Fig 1M). This in line with previous reports indicating that PVT is involved in the CORT response to acute stress [22], but the acute activation of the hypothalamic–pituitary–adrenal axis (HPA) is controlled by neuronal mechanisms other than PVT/CR+ neuronal activity [21,44–46].

These short-term changes demonstrated that POSE could induce behavioural, hormonal, and gene expression alterations consistent with a strong stress exposure. They also show that post-stress photoinhibition of PVT/CR+ neurons is able to alleviate the immediate behavioural changes induced by POSE without influencing the hormonal response to stress.

## Short-term effects of POSE on PVT/CR+ neuronal activity

Next, we aimed to identify how the activity of PVT/CR+ neurons is altered immediately after the post-stress period relative to the pre-stress period. Earlier data indicated decreased GABA-A receptor-mediated inhibition in PVT neurons following restraint stress [47]. Thus, we tested functional GABA-A receptor-mediated synaptic currents after POSE in PVT/CR+ cells using in vitro slice preparations 2 h after POSE (Fig 2A) and compared these to a NOE group. We found that the frequency of spontaneous inhibitory synaptic currents (sIPSCs) significantly decreased in PVT/CR+ neurons (Fig 2B and 2C), while their amplitude remained unaltered (Fig 2D). These data show that the PVT/CR+ cells receive decreased inhibitory synaptic activity immediately after the stress.

To test whether this decreased inhibition can have an immediate consequence on the post-POSE firing activity of PVT/CR+ cells, we used tetrode recordings and optogenetically tagged individual PVT/CR+ neurons. We recorded the same PVT/CR+ activity before the POSE (pre-stress fifth day) and on POST0 day immediately after POSE ($n = 20$ neurons from 4 animals; Figs 2A, 2E, and S3). For these experiments, CR-Cre animals were injected with AAV-DIO-ChR2-EYFP (S3A Fig) which allowed optotagging and clustering (Figs 2E and S4) of PVT/CR+ neurons. We separately analysed firing activity during the wake state outside the nest (wake), wake state inside the nest (nest), and non-rapid eye movement (NREM) sleep inside the nest (sleep).

We found that compared to the PRE5 day activity, on average, PVT/CR+ neurons displayed a significant increase in their firing rate during the nest and sleep states but not during the wake state (Fig 2F–2H) on POST 0 day after the stress exposure. We also analysed the alterations in the firing pattern of PVT/CR+ neurons. Since high-frequency activity (HFA, above 100 Hz) exert disproportionally large effects on their postsynaptic targets and their synaptic plasticity [48,49], we quantified the occurrence of HFA (spike doublets or triplets) in the same PVT/CR+ cell population at PRE5 and POST 0 days. We found that compared to PRE5 day individual PVT/CR+ neurons displayed significantly more HFA clusters in the wake and sleep states (Fig 2I–2K) immediately after the stress.

These data together show that in parallel with the behavioural and hormonal changes PVT/CR+ cell activity is elevated on P0 after the stress event. These data also demonstrate that elevation of c-Fos in PVT/CR+ cells on P0 (Fig 1J and 1K) is paralleled by increased neuronal activity.

## Long-term effects of POSE on spontaneous behaviour

In order to test if POSE is able to persistently alter spontaneous behaviour, we measured and compared the spontaneous wake and sleep behaviours of the EYFP mice during the 5 pre-

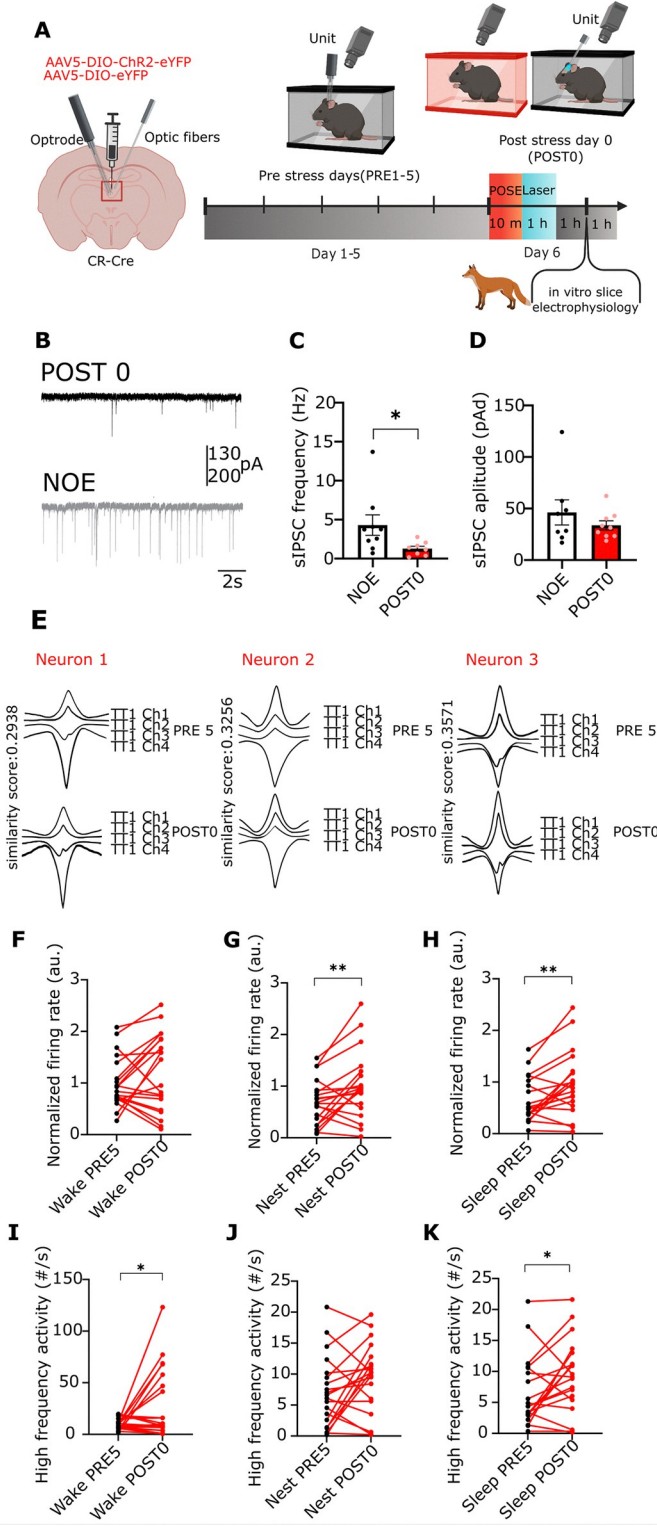

**Fig 2. Short-term changes in PVT/CR+ unit activity after predator odour stress exposure.** (A) Scheme of the experiments representing the in vitro and in vivo recordings. Created with BioRender.com. (B) Example traces of sIPSC recordings from NOE and POSE mice. (C) Bar graphs of average sIPSC frequency (U = 11) and (D) amplitude (U = 30) in PVT/CR+ neurons recorded ex vivo from NOE (*n* = 8 neurons from *n* = 2 mice) and POSE mice 2 h after POSE (*n* = 9 neurons from *n* = 2 mice). (E) Waveforms of 3 optotagged PVT/CR+ neurons recorded by the same

tetrode (TT1). The neurons were recorded for 2 consecutive days (top vs. bottom row). Channel numbers (Ch) and similarity scores (see Methods) between the 2 days are shown. (F–H) Alteration in the firing rate of PVT/CR+ neurons recorded for 1 h both on the PRE5 (black dots) and the POST 0 days after the POSE (red dots) in the (F) wake, (G) nest, and (H) sleep states (F, t[19] = 1.499; G, t[19] = 2.871; H, t[19] = 3.378; $n$ = 20 units from $n$ = 4 mice). The firing rate is normalised to the mean wake population average of the PRE 5 day. (I–K) Same as E–G, for HFA (I, t[19] = 2.463; J, t[19] = 1.847; K, t[19] = 2.411). Underlying data can be found in S2 Data. See S10 Data for the full results of the statistical tests. Data are shown as mean ± SEM. *$p < 0.05$, **$p < 0.01$. HFA, high-frequency activity; NOE, no odour exposure; POSE, predatory odour stress exposure; PVT, paraventricular thalamic nucleus; sIPSC, spontaneous inhibitory synaptic current.

(PRE1-5), and the 4 post-stress days (POST1-4) using intragroup comparison. All measurements were performed in the home cages of the mice (Fig 3A). We found significant alterations in all measures of spontaneous behaviour in the post-stress period (Fig 3). Compared to PRE1–5 days, the horizontal locomotor activity of the EYFP animals remained significantly elevated for the entire duration of the POST period (POST1–4 days) (Fig 3B–3E). After the stress, the animals frequently moved along the edge of their cages (Fig 3C) which was rare in the pre-stress period (PRE period) (Fig 3B). After POSE, the animals needed significantly more time to fall asleep (time between entering the nest and falling asleep, referred to as nest time, Fig 3F) and displayed significantly less nest-building behaviour (Fig 3G and S5 Movie). They increased the time spent with freezing (defined as at least 2 s motionless period) in their nest (Fig 3H and S6 Movie). Both the frequency of the freezing bouts and their total duration increased during the POST period (Fig 3H and 3I). Day-by-day comparison of behavioural data between the PRE and POST periods (RM-ANOVA) also yielded significant differences with the exception of nest building (S5 Fig).

We also measured the quality of sleep by the frequency characteristics (spectral slope) of the EEG (see Methods) [50]. On the stress day (POST0), the spectral slope of the EEG was significantly increased during NREM sleep (S2D Fig) consistent with the earlier data [51,52]. This value, however, did not remain significant during the rest of the post-stress days (POST1–4) indicating normal sleep depth during this period (Fig 3J).

As a control experiment, we compared the behaviour of the NOE group in the PRE and POST period. Our analysis revealed no significant differences in any of the measured behavioural variables during the POST period compared to the PRE period (S6 Fig).

These data collectively demonstrate long-term, stress-induced changes in the spontaneous behaviour of mice within their home cage that occur when the mice are exposed to the predatory odour in a novel environment (see Methods).

## Long-term effects of POSE on PVT/CR+ neuronal activity

We found that the firing activity of PVT/CR+ cells was elevated immediately after POSE (Fig 2E–2K). To investigate whether the neuronal activity of PVT/CR+ cells return to pre-stress levels, as expected, based on the process of homeostatic regulation, or, if it persists for extended periods, we performed long-term recordings of PVT/CR+ neurons. We optotagged $n$ = 69 PVT/CR+ neurons during the PRE1–5 days and $n$ = 57 PVT/CR+ neurons during the POST 1–4 days across 4 animals (Fig 4A) and measured their activity. These cells included those which were used to identify short-term changes in PVT/CR+ firing activity after stress (Fig 2). Some of the neurons ($n$ = 13 tagged cells) could be recorded for up to 3 days (PRE5, POST0, and POST1 days; Figs 4D, 4F, 4H, S3B and S3C).

We found that during the entire post-stress period (POST1–4 days), the mean firing rate of PVT/CR+ cell population did not return to the pre-stress levels. PVT/CR+ neuronal activity remained significantly elevated in all 3 states (wake, nest, and during sleep) (Fig 4B and 4C)

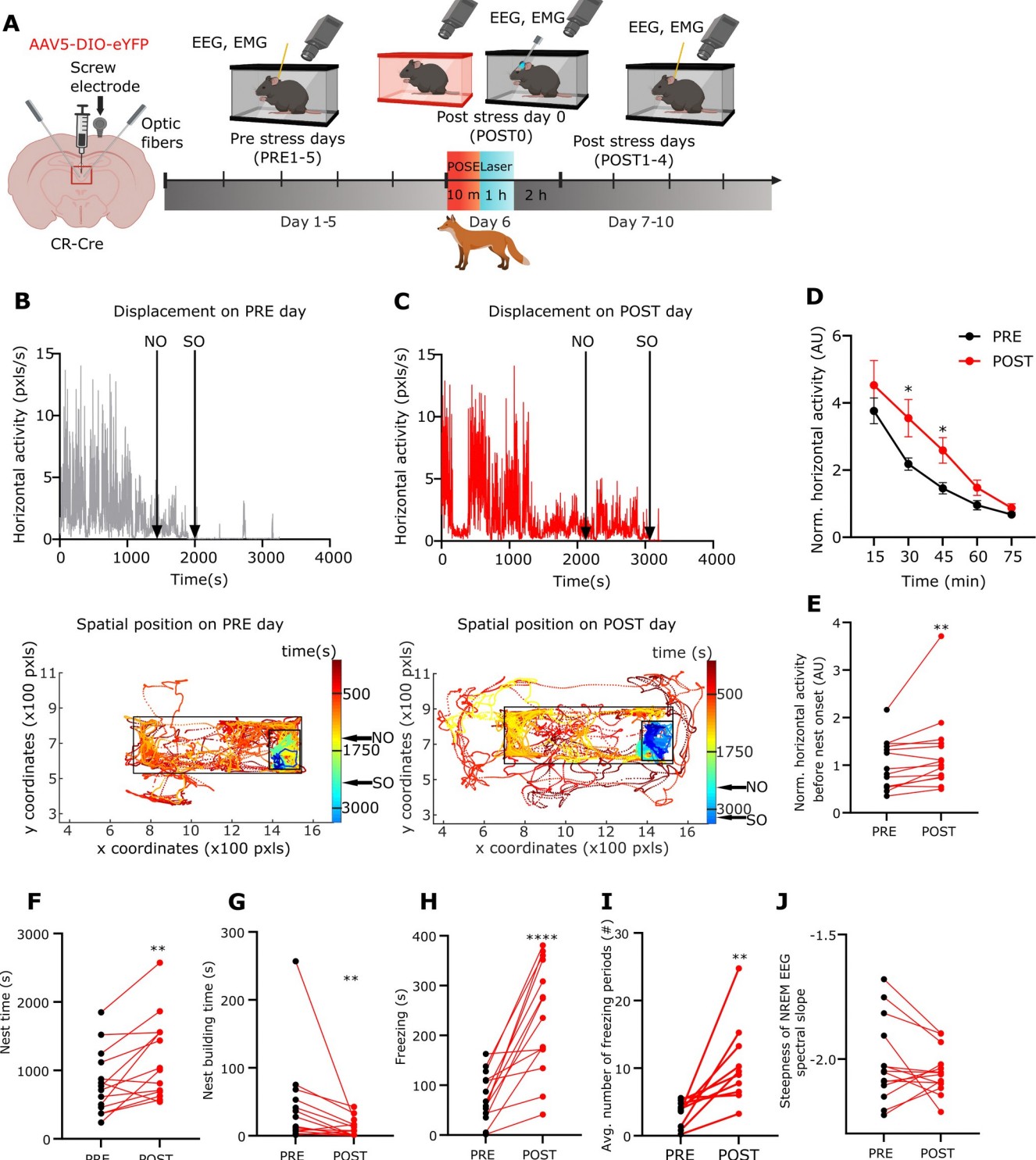

**Fig 3. Long-term behavioural alterations after predatory odour stress exposure.** (A) Scheme of the experiment. Created with [BioRender.com](BioRender.com). (B, C) Top, Representative horizontal locomotor activity during a 3 h session in 1 EYFP mouse within the home cage during one of the PRE days. Nest onset (NO) and sleep onset (SO) are indicated with black arrows. Bottom, Tracked movements of the same animal in the first hour. The dots show the spatial position of the head of the mouse in every frame. Colour of the dots indicate time elapsed (s), from red to blue; small black rectangle marks the nest area; big black rectangle marks cage area. NO and SO, black arrows. (C) Same as B during a POST day. Head positions outside the cage indicate events when the animal was moving along the edge of its cage. (D) Temporal dynamics of averaged, normalised horizontal locomotor activity of EYFP the animals (*n* = 14) during the PRE1–5

days (black) vs. the POST1–4 days (red) periods (30 min, t[13] = 2.438; 45 min, t[13] = 2.392). (E) The averaged, normalised horizontal locomotor activity of EYFP the animals before nest onset (W = 85). Dots represent the averaged daily values of individual animals. (F–J) (F) Nest time (t[13] = 3.579), (G) nest building time (W = −89), (H) freezing time (t[13] = 5.692), (I) number of freezing bouts (t[9] = 3.569) and (J) steepness of the NREM EEG spectral slope (t [13] = 1.086) in EYFP animals (*n* = 14) during the PRE (black dots) and POST (red dots) period. Dots represent the averaged daily values of individual animals. Underlying data can be found in S3 Data. See S10 Data for the full results of the statistical tests. Data are shown as mean ± SEM. *$p < 0.05$, **$p < 0.01$, ***$p < 0.01$, ****$p < 0.001$. NREM, non-rapid eye movement.

during the entire post-stress period. As indicated above (Fig 2F–2H) during the pre-stress period PVT/CR+ neurons displayed state-dependent activity. Firing rate was highest during wake, lower in the nest and lowest during non-REM sleep (Fig 4C). In order to test whether increased firing rate observed after the stress affects all the 3 states to a similar degree, we calculated state modulation indices (SMI, see Method) for each pair of states (wake-to-nest, nest-to-sleep, wake-to-sleep) for every recorded PVT/CR+ cells for the PRE and POST periods. Compared to the pre-stress period after the stress PVT/CR+ neurons displayed significantly less state modulation in the wake-to-nest and wake-to-sleep relations as shown by their lower SMI values (Fig 4D and 4E). Less state modulation was largely due to a larger post-stress increase in firing rates recorded during the resting states (nest and sleep) relative to the wake state (Fig 4C).

Next, we analysed whether alteration in firing patterns can also be demonstrated during the post-stress period. We found that relative to the PRE period the occurrence of HFA remained elevated in all 3 behavioural states during the POST1–4 days (Figs 4F, 4G, S7A and S7B). Next, we analysed the correlated activity (CA, spikes of 2 simultaneously recorded neurons within 5 ms) among simultaneously recorded PVT/CR+ cells. Similar to HFA, we found that the occurrence of CA also significantly increased in all 3 states (Fig 4H and 4I). After appropriate normalizations of HFA and CA with the background firing activity (see Methods), these changes remained significant only in the nest CA values (S7C and S7D Fig). This indicates that a general increase of the mean firing rate in PVT/CR+ neurons is largely responsible for the elevated occurrence of HFA and CA.

These data together show that the activity PVT/CR+ cells display state-dependent changes persisting for days, after a single stress exposure paralleling the observed behavioural changes. Neurons postsynaptic to PVT/CR+ cells receive more HFA and CA from the PVT/CR+ cells for several consecutive days.

## Effect of post-stress photoinhibition on long-term changes in behaviour

Prolonged alteration of behaviour in the post-stress period (Fig 3) correlated with persistent, stress induced changes of PVT/CR+ neuronal activity (Fig 4). Thus, next, we asked whether reducing the activity of PVT/CR+ cells for 1 h after the stress event can prevent long-term changes in the behaviour during the POST1–4 days as well (Fig 5A). We found that post-stress inhibition on the day of the stress was able to prevent the development behavioural alterations during the entire post-stress period. Compared to the PRE 1–5 days SwiChR mice exhibited no significant, stress-induced behavioural changes during the POST1–4 days. Locomotion decreased, not increased after the stress (Fig 5B–5E). There was no alteration in the nest time (Fig 5F) or in the nest building (Fig 5G). Frequency of freezing bouts and their duration did not change during the POST period relative to the PRE period (Fig 5H and 5I). Day-by-day comparison of behavioural data between the PRE and POST periods (RM-ANOVA) yielded no significant differences (S8 Fig). Post-stress inhibition, however, significantly, increased the NREM EEG spectral slope during the POST period (Fig 5J) indicating a prolonged effect on sleep.

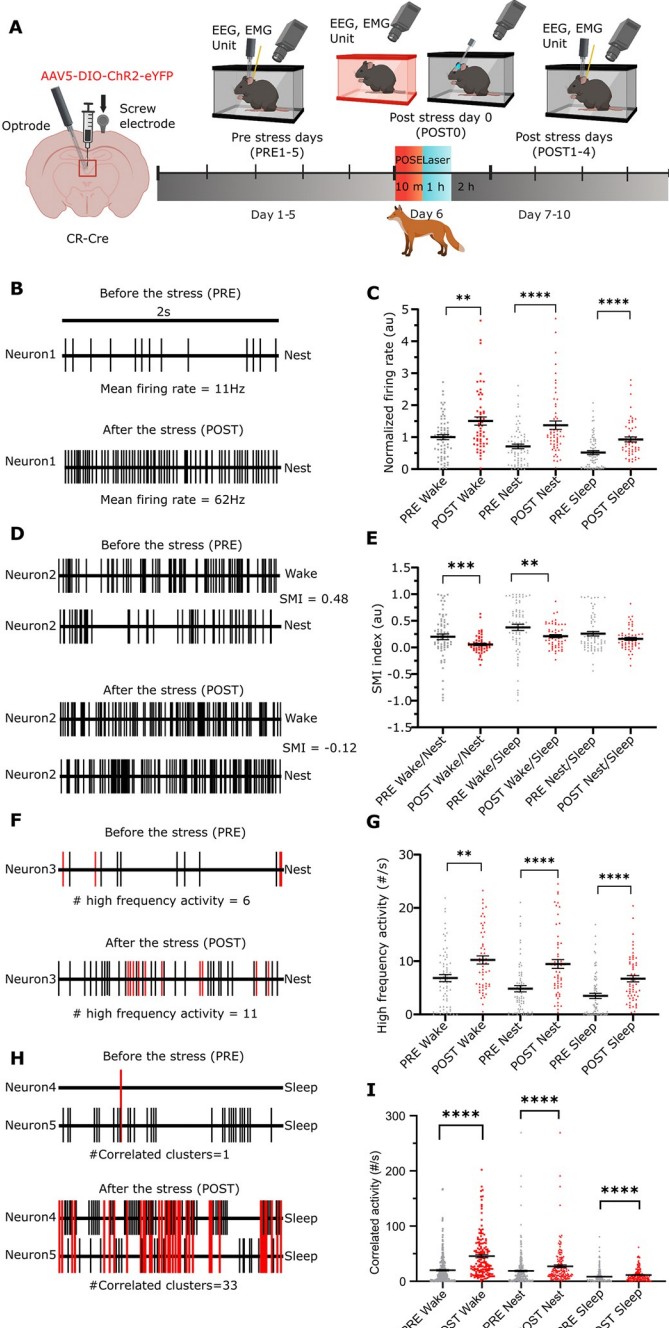

**Fig 4. Long-term changes of PVT/CR+ unit activity after predator odour stress exposure.** (A) Scheme of the experiment. Created with BioRender.com. (B) Change in the firing rate of an example PVT/CR+ neuron recorded on PRE 5 day and POST 1 day. (C) Mean firing rate values of PVT/CR+ neurons ($n$ = 126 neurons in 4 animals) during wake, nest, and sleep during the PRE 1–5 and the POST 1–4 days periods. Dots represent individual cells. The data are normalised to the mean wake firing rate of the PRE period (wake, U = 1357, $p$ = 0.0026; nest, U = 1,041, $p < 0.0001$; sleep U = 1,097, $p < 0.0001$). (D) Alteration in the state-dependent activity of an example PVT/CR+ neuron recorded on PRE 5 and POST 1 days. (E) SMI in the entire population of PVT/CR+ neurons during PRE and POST periods (wake/nest SMI, U = 1,295, $p$ = 0.0009; wake/sleep SMI, U = 1,304, $p$ = 0.0011; nest/sleep SMI, U = 1,814, $P$ = 0.4576). (F) Change in HFA of an example PVT/CR+ neuron recorded on PRE 5 and POST 1 days. Red lines (spikes) show HFAs, black lines mark the spikes with longer interspike intervals than 10 ms. (G) Frequency of HFAs in the entire PVT/CR+ population during the entire PRE and POST periods (wake, U = 1,309, $p$ = 0.0011; nest, U = 992, $p < 0.0001$; sleep U = 1,034, $p < 0.0001$). (H) Alteration in CA between a representative pair of PVT/CR+ neurons recorded on PRE 5 and POST 1 days. Linked red lines (spikes) indicate CAs, black lines mark the spikes occurring

outside the time window of 5 ms. (I) Frequency of CA in the entire population of PVT/CR+ cell pairs ($n$ = 448 pairs) during the PRE and POST period. (U = 13,625, $p < 0.0001$; nest, U = 2,0351, $p < 0.0001$; sleep U = 20,049, $p < 0.0001$). Underlying data can be found in S4 Data. See S10 Data for the full results of the statistical tests. Data are shown as mean ± SEM. **$p < 0.01$, ***$p < 0.001$, ****$p < 0.0001$. CA, correlated activity; HFA, high-frequency activity; PVT, paraventricular thalamic nucleus; SMI, state modulation index.

As a control, we performed post-stress photoinhibition in the NOE group and compared their behaviour in the PRE and POST period. This manipulation resulted in decreased nest time, reduced freezing behaviour and increased nest-building activity (S9 Fig). Among these behaviours, the changes in freezing and nest-building activity were statistically significant.

These data indicate that post-stress photoinhibition immediately after the stress event has prolonged effects on stress-induced behaviour. Interference with PVT/CR+ neuronal activity, even in the absence of predatory odour, can significantly reduce stress-related behaviours for several days.

## Effect of post-stress photoinhibition on long-term changes in PVT/CR+ neuronal activity

Next, we investigated whether post-stress inhibition of PVT/CR+ cells for 1 h can also prevent the long-term alterations in the firing activity of PVT/CR+ cells (Fig 6A) for the entire POST period similar to the behavioural changes. For these experiments, we used SwiChR injected animals implanted with tetrodes. We found that the mean average firing rate of PVT/CR+ neurons in the SwiChR animals calculated for the entire POST period (POST1–4 days) did not significantly change ($n$ = 39 neuron in 2 mice) in any state relative to the PRE period (PRE1–5 days, $n$ = 38 in 2 mice) (Fig 6B). The marked post-POSE alteration in the state-dependent firing, observed in EYFP animals (Fig 4E) was also abolished, thus state-modulation of neuronal activity did not decrease during the post-POSE period (Fig 6C). In fact, in the case of wake/sleep SMIs PVT/CR+ neurons increased not decreased their state modulation indices after stress in the SwiChR group (Fig 6C). Post-POSE HFA activity (Figs 6D and S10A) and CA of PVT/CR+ cells also remained unaltered during the nest and sleep state (Fig 6E). We found only one measure, CA during the wake state, which increased significantly compared to the PRE period similar to the EYFP group (Figs 6E and S10B).

These data show that inhibiting the activity of PVT/CR+ cells immediately after stress was able to prevent not only the emergence of stress-induced behavioural consequences but also the alteration of the firing activity in PVT/CR+ neurons for the entire POST period as well.

## Role of PVT/CR+ neurons in stress-induced changes of their forebrain targets

Next, we aimed to identify the impact of altered post-stress firing of PVT/CR+ neurons on their forebrain targets by measuring changes c-Fos expression in the EYFP ($n$ = 7 animals) and SwiChR ($n$ = 6 animals) groups (Figs 7A and S11) 1 h after the POSE. We also included home cage ($n$ = 7) and no odour ($n$ = 5) controls in these experiments. We selected 3 postsynaptic regions in which significant proportions of the subcortical glutamatergic inputs arise from PVT/CR+ cells (prelimbic cortex (PrL), basolateral amygdala (BLA), and shell of the nucleus accumbens (NAcS)) (Fig 7B) [13] and another two, the central amygdala (CeA) and bed nucleus of stria terminalis (BNST) to which, substantial glutamatergic inputs arrive from subcortical centres other than PVT [53]. Since PVT/CR+ neurons were labelled in these animals, we specifically quantified c-Fos positive neuron numbers in those downstream areas within these structures that contained labelled PVT/CR+ axons, hence were under direct excitatory

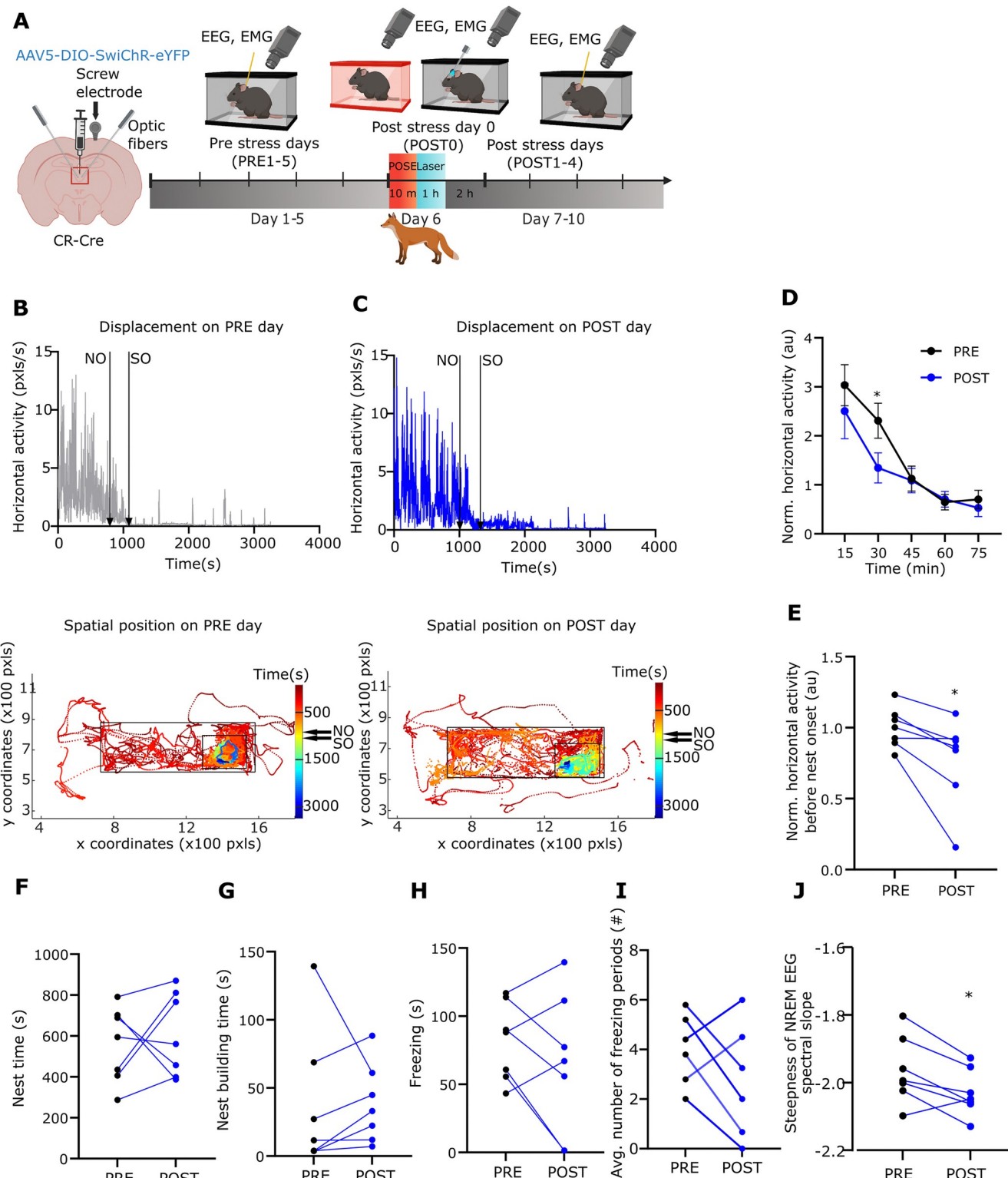

**Fig 5. Post-stress photoinhibition of PVT/CR+ neurons prevents stress induced, long-term behavioural changes.** (A) Scheme of the experiment. Created with BioRender.com. (B) Top, Changes in horizontal locomotor activity during 3 h in one SwiChR mouse within the home cage on a PRE day. Nest onset (NO) and sleep onset (SO) are indicated with black arrows. Bottom, Tracked movements of the animal in the first hour. The dots show the spatial position of the head of the mouse in every frame and the colour of the dots indicate lasting time (s), from red to blue; small black rectangle marks the nest area; big black rectangle marks cage area. Nest onset (NO) and sleep onset (SO) are indicated with black arrows. (C) Same as (B) on a POST day. (D) The dynamics of

averaged, normalised horizontal locomotor activity (E) of SwiChR mice (*n* = 7) during the PRE (black) vs. the POST (blue) periods (t[6] = 2.835). (E) Averaged, normalised horizontal locomotor activity of SwiChR mice during the PRE (black) vs. the POST (blue) periods (t[6] = 2.944). (F) Nest time (t[6] = 0.4886), (G) nest building time (t[6] = 0.1084), (H) freezing time (t[6] = 1.142), (I) number of freezing bouts (t[5] = 1.347), and (J) steepness of NREM EEG spectral slope (t[6] = 2.969) in SwiChR mice during the PRE and POST periods. Dots represent the average daily values of individual animals. Underlying data can be found in S5 Data. See S10 Data for the full results of the statistical tests. Data are shown as mean ± SEM. *$p < 0.05$. NREM, non-rapid eye movement; PVT, paraventricular thalamic nucleus.

PVT influence (Fig 7B). We found that compared to home cage and NOE controls POSE markedly increased c-Fos expression in all investigated downstream brain regions except the BNST (Fig 7C–7G). Post-POSE photoinhibition of PVT/CR+ neurons significantly reduced c-Fos expression in the PrL (Fig 7C), BLA (Fig 7D), and NAcS (Fig 7E), but not in the CeA (Fig 7F) and BNST (Fig 7G). This is consistent with the fact that most subcortical excitatory inputs to PrL, BLA, and NAcS arise in PVT. SwiChR inhibition of PVT/CR+ cells in these regions alleviates stress-related elevation of excitatory activity resulting in lower c-Fos expression (Fig 7C–7G). In contrast, CeA and BNST receive other major subcortical glutamatergic inputs [53], which can transmit stress signals independently of PVT/CR+ input, allowing c-Fos expression to persist despite reduced PVT/CR+ activity (Fig 7C–7G).

These data illustrate that altered PVT/CR+ activity immediately after the stress can lead to increased c-Fos activity in their postsynaptic targets. Reducing the activity of PVT/CR+ neurons is able to alleviate the stress-induced molecular activation in key forebrain regions which control adaptive behaviours.

## Stress-induced changes in homeostatic GABA-A receptor expression

We observed a prolonged increase in the firing activity of PVT/CR+ neurons (Fig 4). Increased activity is linked to a compensatory, homeostatic adjustments in synaptic inhibition [54]. Thus, using high-resolution immunocytochemistry we tested how the expression levels of α1 and γ2 subunits of synaptic GABA-A receptor proteins are altered in PVT/CR+ neurons as a result of the elevated post-POSE activity on POST1 (Figs 7A and S11) in EYFP (*n* = 3), SwiChR (*n* = 4), and home cage control (*n* = 3) groups (Fig 7H). It is known that γ2 subunits exhibit primarily synaptic localization, whereas α1 subunits, which co-assemble with γ2 to form functional receptors are found both synaptically and extrasynaptically [55,56]. We found that the expression of both GABA-A receptor subunits significantly increased in the EYFP animals after POSE compared to home cage controls (Fig 7I–7K), suggesting a homeostatic up-regulation of GABA-A receptors potentially evoked by the increased firing of PVT/CR+ neurons 1 day after the stress event. Similar to other post-stress changes post-POSE SwiChR activation was able to prevent these alterations in α1 and γ2 expression (Fig 7I–7K). These data suggest a homeostatic up-regulation of GABA-A receptor expression in PVT/CR+ cells 24 h after the stress which is absent in the inhibited group.

## Effects of late inhibition of PVT/CR+ cells on the stress-induced behavioural consequences

The data above demonstrate that the altered activity of PVT/CR+ neurons immediately after the stress have a critical role in establishing stress-induced alteration of behaviour for several days. It is still unclear, however, whether PVT/CR+ neurons are actively involved in the maintenance of stress-induced phenotype at later time points, or, alternatively later during the post-stress period other brain regions take over this role. To gain insight into this question, we applied a 1-h bout of photoinhibition not immediately after POSE but 5 days later (LATE group; Figs 8A and S12), when mice still exhibited robust stress-induced behavioural

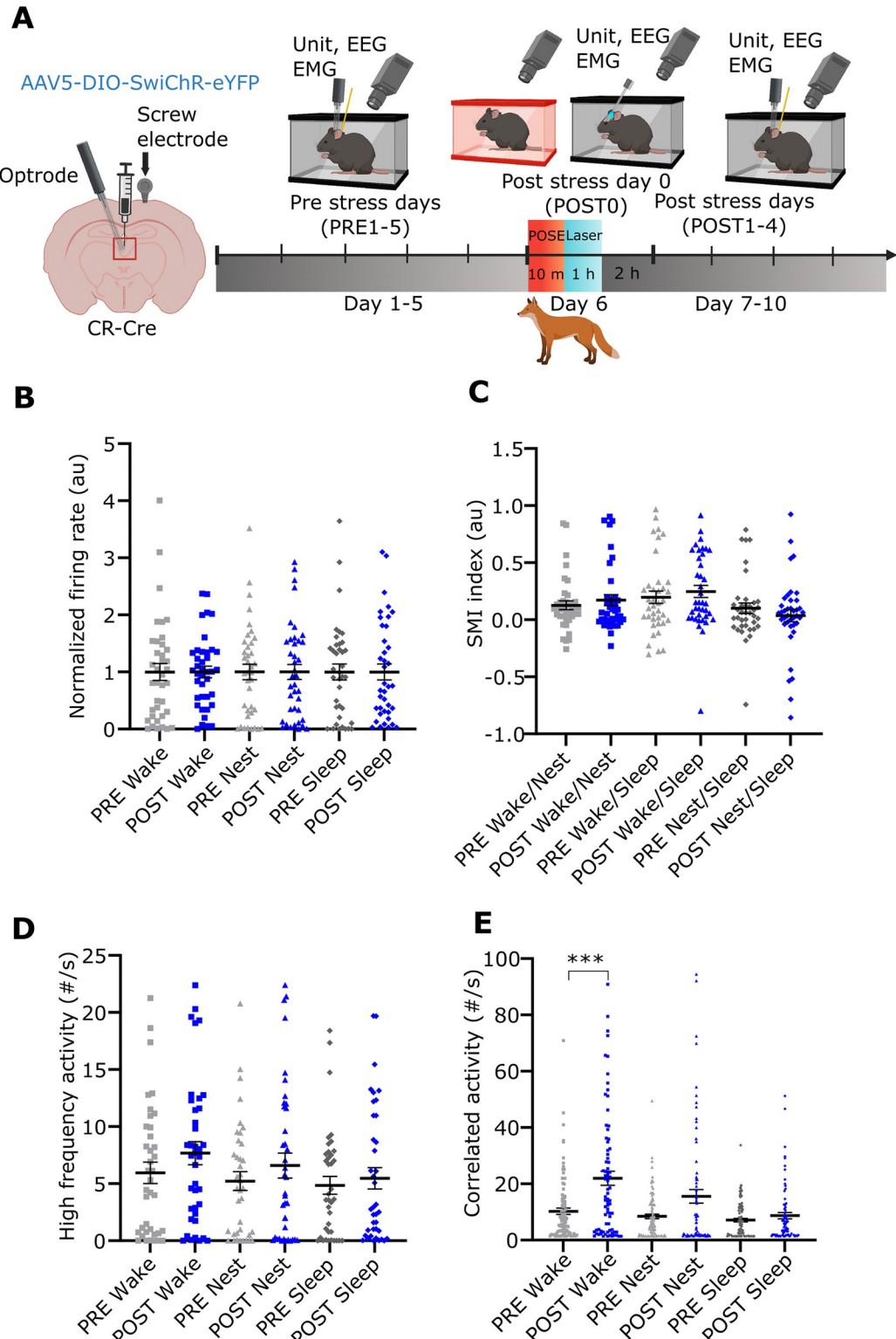

**Fig 6. Effect of post-stress photoinhibition on long-term changes in PVT/CR+ neuronal activity.** (A) Scheme of the experiment. (B) Post-POSE photoinhibition of PVT/CR+ neurons on POST0 day prevents alterations in the firing rate, (C) SMI, (D) occurrence of HFA, and (E) CA during the PRE (black, $n = 38$ neurons in 2 mice) vs. the POST (blue, $n = 39$ in 2 mice) periods (all $p > 0.05$ unless indicated otherwise). Underlying data can be found in S6 Data. See S10 Data for the full results of the statistical tests. Data are shown as mean ± SEM. *$p < 0.05$, ***$p < 0.001$. CA, correlated activity; HFA, high-frequency activity; PVT, paraventricular thalamic nucleus; SMI, state modulation index.

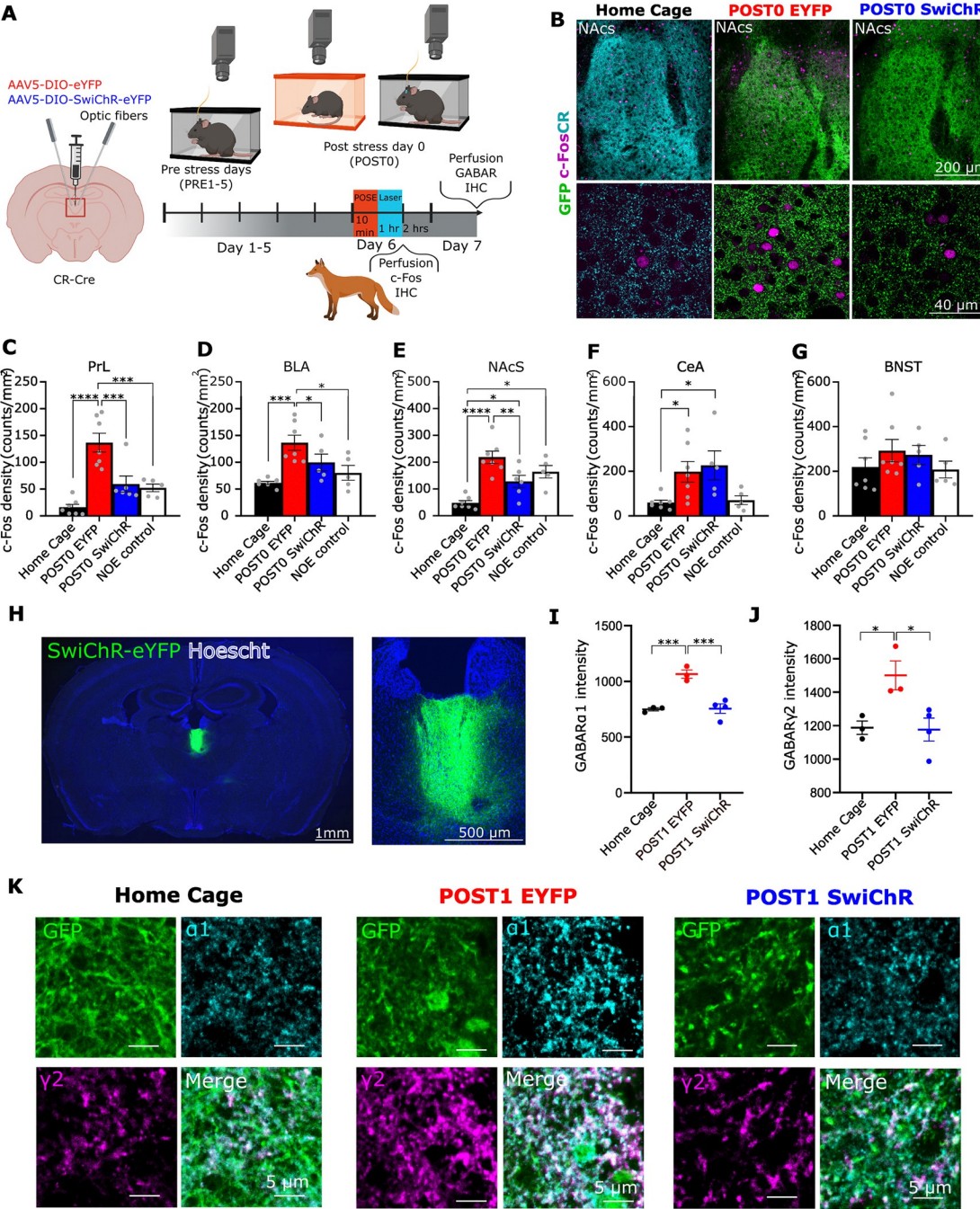

**Fig 7. Role of PVT/CR+ neurons in stress-induced molecular changes.** (A) Scheme of the experiments representing the timeline for GABA-A receptor and c-Fos immunostainings. Created with BioRender.com. (B) Top, Representative, low power (upper panel) and high power (bottom panel) confocal images showing c-Fos expression (magenta) in the nucleus accumbens shell (NacS) from Home Cage control, EYFP, and SwiChR mice. Cyan (left) or green (middle and right) indicates CR- or GFP immunostaining, respectively, labelling PVT axons in NacS. (C–G) Quantification of c-Fos positive neurons in (C) PrL, (D) BLA, (E) NacS, and (F) CeA, (G) BNST in Home Cage ($n = 7$), EYFP ($n = 7$), SwiChR ($n = 6$), and NOE ($n = 5$) mice on POST0 day (PrL, $F_{(3,21)} = 17.10$, $p < 0.0001$; BLA, $F_{(3,21)} = 7.503$, $p = 0.0013$, NacS, $F_{(3,21)} = 14.49$, $p < 0.0001$; CeA, $F_{(3,20)} = 4.608$, $p = 0.0073$; oval nucleus of the BNST $F_{(3,20)} = 0.848$, $p = 0.4837$). (H) Representative low and high power fluorescent images from the PVT of AAV5-DIO-SwiChRCA-EYFP injected mice. Blue, Hoechst counterstaining. (I, J) Changes in the fluorescence intensity of (H) GABA-A α1 and (I) GABA-A γ2 immunostainings on POST1 day (GABA-A α1, $F_{(2,7)} = 23.08$, $p = 0.0008$; GABA-A γ2, $F_{(2,7)} = 6.692$, $p = 0.0237$; Home Cage $n = 3$ mice; EYFP $n = 3$ mice; SwiChR $n = 4$ mice). (K) Representative confocal images showing GABA-A α1 and GABA-A γ2 immunostainings in the PVT from Home Cage control, EYFP, and SwiChR mice on POST1 day. Underlying data can be found in S7 Data. See S10 Data for the full results of the statistical tests. Data

are shown as mean ± SEM. *$p < 0.05$, **$p < 0.01$, ***$p < 0.001$, ****$p < 0.0001$. BLA, basolateral amygdala; BNST, bed nucleus of the stria terminalis; CeA, central amygdala; NAc, nucleus accumbens; NOE, no odour exposure; PrL, prelimbic cortex; PVT, paraventricular thalamic nucleus.

phenotype and altered PVT/CR+ firing activity. We examined spontaneous behaviour for 5 more days after the late inhibition.

Similar to P0 photoinhibition, late inhibition had an immediate effect on locomotor activity (Fig 8B) but late inhibition was also able to induce prolonged effects for the entire late period (POST 5–9 days). Compared to the POST1–4 days following the late inhibition (POST 5–9 days), the horizontal activity (Fig 8C and 8D) and freezing behaviour (Fig 8E) were significantly reduced. Day-by-day comparison of the data between the POST1–4 and POST 5–9 periods (RM-ANOVA) also yielded significant differences (S13 Fig). In contrast, nest time (Fig 8F) and nest building time (Fig 8G) were not significantly affected.

These observations suggest that 5 days after the stress the activity of PVT/CR+ cells still contribute to the maintenance of the stress-induced behavioural profile. They also demonstrate that similar to the P0 intervention, the effect of a single bout of photoinhibition in PVT/CR+ cells can last for several days. In contrast to the P0 group, however, not all stress-induced changes were ameliorated in the LATE cohort.

## Discussion

Our present data show that PVT/CR+ cells do not simply transfer stress-related signals to their forebrain targets but undergo a robust, persistent increase in firing activity, lasting for several days, in response to a strong inescapable stressor. To our knowledge, no such a long-term increase in firing rate has been described in the brain, so far. This long-term increase of PVT/CR+ neuronal activity had a causal role in establishing the alteration of spontaneous behaviour after stress. Inhibition of PVT/CR+ neurons after the stress event, for only 1 h, could prevent the emergence of stress induced alteration of PVT/CR+ activity as well as the development of stress-induced behaviour during the post-stress days. Our optogenetic approach allowed us to selectively inhibit PVT/CR+ neuronal activity specifically after, but not during, the stress exposure. This precise temporal control ensured that we did not interfere with PVT/CR+ activity during the presentation of the stressor. This experimental approach preserved unaltered perception of the stress event in the inhibited group. The data together suggests that the activity of PVT/CR+ cells in the early post-stress period is critical for the establishment of behavioural alterations induced by stress.

In this study, the stressor was an inescapable threat (predator odour) based on innate fear in novel environment (see Methods). In agreement with the literature, we found strong stress response to 2MT as demonstrated by robust defensive behaviour during and immediately after the exposure, elevated c-Fos expression in PVN, and a significant increase in blood corticosterone levels after the POSE [36,39,40,42]. The data are also consistent with earlier results showing strong stress evoked activity in PVT cells [19,20,57,58].

Based on these data, we propose the following neuronal mechanism for the development of acute stress induced behavioural alterations. A single exposure to a potent stressor results in a sustained elevation of PVT/CR+ neuronal activity. Excessive excitation in PVT/CR+ cells is likely brought about by the high number of subcortical inputs it receives [13,33,34]. As a result of the profusely branching collaterals of PVT/CR+ cells [13], the altered activity is transmitted to their forebrain targets (BLA, CeA, PrL, HIPP, BNST, and NAc) which have well established for their involvement in defensive behaviour [13,33,34,59] as a synchronous glutamatergic signal. Indeed, we found that the stress-induced c-Fos activity in those forebrain targets of PVT/

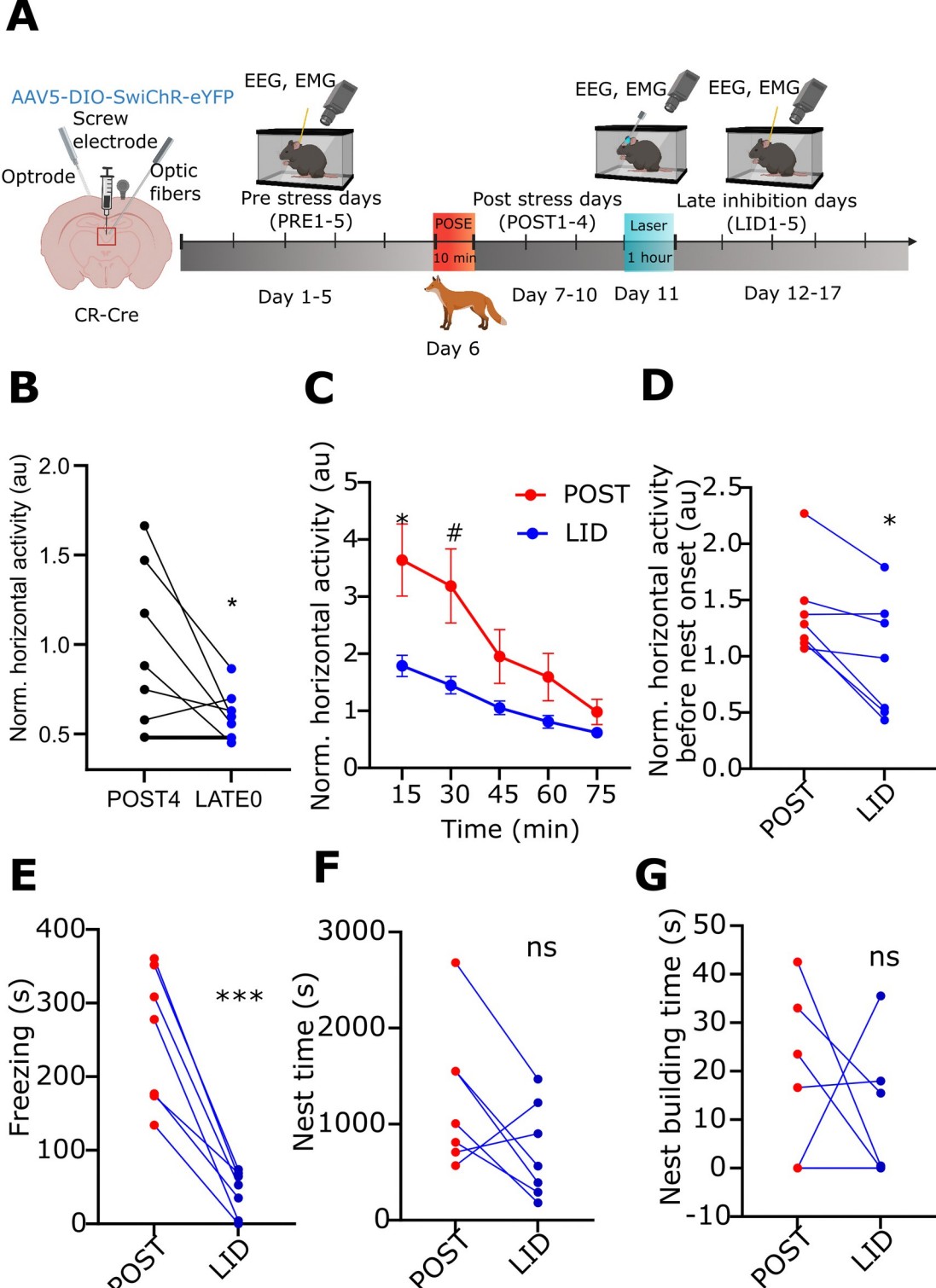

**Fig 8. Effect of late inhibition of PVT/CR+ neurons on stress-induced behavioural changes.** (A) Scheme of the experiment. Created with BioRender.com. (B) Averaged, normalised horizontal activity during the photoinhibition in LATE0 day compared to the preceding day (POST4) (t[6] = 2.474, (*n* = 7)). (C) Dynamics of averaged, normalised horizontal locomotor activity of SwiChR mice during the POST (POST 1–4 days, red) vs. the LID (LID 1–5 days, blue) periods (15 min, t[6] = 2.738, *p* = 0.0338; 30 min, t[6] = 2.443, *p* = 0.0503). (D) Averaged, normalised horizontal locomotor activity of SwiChR mice during the POST (POST 1–4 days, red) vs. the LID (LID 1–5 days, blue) periods (W = −26). (E) Freezing behaviour (t[6] = 6.909, *p* = 0.0005), (F) nest time (t[6] =

2.031), and (G) nest building time (W = −5) in SwiChR mice during the POST (POST 1–4 days, red dots) and LID (LID 1–5 days, blue dots) periods. Dots represent the average daily values of individual animals. Underlying data can be found in S8 Data. See S10 Data for the full results of the statistical tests. Data are shown as mean ± SEM. Ns means no significant difference between groups, #p < 0.06, *p < 0.05, ***p < 0.001. PVT, paraventricular thalamic nucleus.

CR+ neurons (PrL, Amy, and NAc) which receives the majority of their subcortical inputs from PVT is diminished if PVT/CR+ neurons are inhibited after the stress. This is consistent with the facts that PVT/CR+ cells are able to evoke strong postsynaptic response in these brain areas [13] and that PVT/CR+ neurons persistently increased their activity after the stress exposure (present study). We found that the high-frequency activity (<100 Hz) of PVT/CR+ was also significantly elevated after the stress. Since these events have a large impact on the postsynaptic targets [48,49], they have the capacity to induce morphological and functional plasticity in the forebrain known to accompany stress-induced changes [1,2,60].

Our data highlight the sensitivity of pre-sleep behaviour to stress. Prior to the stress exposure, the animals displayed normal nest building, spent little time freezing in the nest, and fell asleep relatively fast after they entered the nest. During the pre-stress days, the firing rate of PVT/CR+ cells significantly slowed down as the animals entered the nest and performed pre-sleep behaviours and it was further reduced as the animal fall asleep. These behavioural and physiological data support the view that the wake state in the nest is a transient arousal state between wake and sleep and represents a preparation to sleep. Following the stress exposure, both the behaviour and the firing activity in the nest changed significantly. Mice exhibited heightened freezing behaviour, diminished nest-building activities, and increased latency to fall asleep. In parallel with these behavioural changes, the firing rate of PVT/CR+ cells did not drop in the nest relative to wake state following the stress event. These results indicate that, similar to humans, exposure to a stress event can have profound impact on pre-sleep behaviour. Our data demonstrate that the increase in PVT/CR+ neuronal activity within the nest after exposure to stress plays a causal role in the observed alterations in nest-related behaviour.

We found that PVT/CR+ neuronal activity affects behaviour immediately after the stress unlike in a previous study that showed a delayed role of PVT (after 24 h) [16] in a conditioned fear paradigm. We think that these data are not mutually exclusive. We measured the effect of PVT activity on spontaneous behaviour, whereas Do-Monte and colleagues [16] examined the role of PVT in the retrieval of a stressful event. These functions likely operate through distinct mechanisms. Our data indicate an immediate bottom-up effect of PVT on the forebrain mediating alterations of spontaneous behaviour in response to stress, while Do-Monte and colleagues studied a delayed, top-down cortical influence on PVT in memory retrieval.

Sustained increase in firing activity is an infrequent observation within the brain. The firing rates of individual neurons are maintained by a cell-autonomous form of homeostatic plasticity (called "synaptic scaling") [4–6,8]. In our study, sustained increase persists for several days, representing a novel and, to our knowledge, unprecedented finding. Mechanisms governing of long-term changes in neurons within the fear circuit are actively under investigation [61]. Abundant in vitro data demonstrate that PVT neurons are endowed with a range of intrinsic features that allows them to alter their membrane properties as a result of various hormonal, diurnal, or stress factors [18,62]. These include potential contributions from $Ca^{2+}$ activated $Ca^{2+}$ channels [63] GIRK-like and TRPC-like conductuctancies [64], alteration in leak potassium, $I_h$ low- and high-threshold $Ca^{2+}$ currents [65–67], all of which might contribute to the elevated firing rate observed in this study. Additionally, our data indicates a reduction in synaptic inhibition immediately following stress exposure, aligning with previous findings [47]. It is important to note that these initial changes likely precede long-term network alterations,

including top-down cortical influence [16] as evidenced by the continued increase in PVT/CR+ firing activity in the later post-stress days. These results together emphasises the complexity of the dynamic cellular and network processes issued within the fear circuit following exposure to stress. For how long elevated PVT/CR+ are maintained after the stress event and what are the mechanisms that may return their activity to the pre-stress level are presently unclear and remain to be investigated.

Optogenetic inhibition of PVT/CR+ cells resulted in a marked reduction of firing rate during the laser ON periods. The activation of SwiChR demonstrated a significant reduction in the stress-induced increase in c-Fos activity in PVT/CR+ cells, confirming the inhibitory effect of the opsin. Our measurements on the spatial extent of reduced c-Fos expression showed that the effect of photoinhibition did not extend beyond than 500 μm from the tip of the optic fibres, effectively ruling out "off-target" effects. The attenuation of PVT/CR+ neuron activity for 1 h after the stress event exerted a prolonged influence on both PVT/CR+ activity and behaviour. The optogenetic interference with post-stress neuronal activity was able to block both the shift in the firing regime of PVT/CR+ induced by stress for the whole duration of observations (5 days) and the emergence of behavioural disturbances. These data indicate that the optogenetically evoked Cl⁻ entry via the SwiChR molecule during the early post-stress period can counteract the mechanisms that eventually lead to the prolonged stress-induced shift in PVT/CR+ excitability. These results underscore the central role of PVT/CR+ cells in post-stress forebrain activity. The absence of an increase in firing rate in the inhibited group during POST1-POST4 days indicates that, in the absence of the initially altered post-stress PVT activity, no other brain centre upstream to PVT could take on the role of driving elevated PVT/CR+ neuronal activity and altered behaviour. Recent data on the role of PVT neuronal ensembles which encode early-life-adversity in adult reward behaviour indicate that altered PVT activity may persist from immature to adult stage [68].

Increased neuronal activity is known to lead to a homeostatic up-regulation of GABA-A receptor density [54]. Accordingly, we also found that in parallel with the elevated unit activity, expression of synaptic GABA-A receptors subunits was significantly increased in PVT/CR+ cells after the stress. Apparently, however, in the EYFP group the increased synaptic GABA-A receptor expression was not sufficient to reduce the elevated post-stress firing rate of PVT/CR+ cells. Moreover, since GABAergic mechanisms are known to be involved in the synchronization of neuronal activity [69,70], this homeostatic increase in inhibition can participate in the increased synchrony among PVT/CR+ cells observed in the post-stress period. The post-stress up-regulation of GABA-A receptors also did not take place in the SwiChR group probably due to the fact that in this group the mean firing rate remained unaltered after the stress.

Inhibition of PVT/CR+ neurons 5 days after stress was still able to alleviate some (but not all) stress-induced behavioural alterations. The effectiveness of this late behavioural response is rather surprising. However, it is in line with our observations that PVT/CR+ neuronal activity remains high even on POST 5 day. The data suggest that similar to POST 0 photoinhibition interference with PVT/CR+ activity at POST 5 can re-normalise PVT/CR+ activity and the associated behaviours. On the other hand, the data also shows that after 5 days the maintenance of not every post-stress behavioural change depend on the persistently elevated PVT/CR+ activity. Some changes, such as increased nest time and decreased nest building, to appear have already consolidated in the forebrain targets due to prolonged excitatory bombardment of PVT/CR+ cells. These stress-induced changes are maintained even if PVT/CR+ activity returns to normal.

## Conclusions

A single exposure to a stressful event can lead to the emergence of anxiety and trauma-related disorders [71]. In this study, using a rodent model of acute stress exposure, we aimed to decipher a mechanism that can mediate the development of acute effects of stress on behaviour. The critical observation is that PVT/CR+ cells can act as a bidirectional tuneable device. Transient interference with their activity either in the positive (POSE) or negative (SwichR) direction has prolonged effects on their firing regime lasting for days. Through their widespread forebrain connections altered PVT/CR+ activity can have profound, long-lasting effects on both promoting and reducing stress-related behaviours. The similarity in the organisation of PVT and their inputs between mice and humans [13] and the involvement of paramedian thalamus in arousal-related problems [72,73] suggest that PVT may play a similar role in humans. The reversible plastic alteration of PVT/CR+ neurons after the stress may offer a window of opportunity to treat disorders related to acute stress exposure.

## Methods

All procedures with mice were approved by the Animal Welfare Committee of the Institute of Experimental Medicine, Budapest, conformed to guidelines established by the European Community's Council Directive of November 24, 1986 (86/609/EEC). Experiments were authorised by the National Animal Research Authorities of Hungary (PE/EA/877-7/2020).

### Subjects

Adult male *Calb2-IRES-Cre* (CR-Cre, C57Bl/6J, *n* = 73) mice (postnatal 3 to 5 months) were used in the experiments from the Jackson Laboratory (RRID: IMSR_JAX:10774). All animals were group-housed (3 to 4/cage) in Plexiglas chambers at constant temperature (22 ± 1°C) and humidity (40% to 60%) under a circadian light-dark cycle (lights-on at 9 AM, lights-off at 9 PM). All mice were single housed after electrode and optical fibre implantation. For the animals, the food (Sniff) and water were provided ad libitum.

### Viral vectors

For control mice in the behavioural, sleep, corticosterone, c-Fos, and GABA-receptor localization experiments, we used AAV5.EF1a.DIO.eYFP.WPRE.hGH (Addgene27056) vector (EYFP mice), whereas for optogenetic inhibition in the same experiments, AAV5.EF1a.DIO. SwiChRCA.TS.EYFP.WPRE (UNC Vector Core) vector was injected (SwiChR mice). In the optrode experiments for the not inhibited, control animals (*n* = 4 ChR2 animals), we utilised AAV5.EF1a.DIO.hChR2(H134R)-eYFP.WPRE.hGH (Addgene20298P) vector for optical tagging, whereas for the inhibited animals with unit recordings, the same SwiChR construct was utilised to phototagging as above (*n* = 2 SwiChR animals).

### Surgery

Mice were deeply anesthetised with ketamine-xylazin (intraperitoneal injection, ketamine, 83 mg/kg; xylazine, 3.3 mg/kg). Depth of sleep was monitored throughout the surgery, if needed supplemental dose (ketamine, 28 mg/kg; xylazine, 1.1 mg/kg) was injected intramuscularly. Externally applied lidocaine was used on the head and ears. The mice were placed in a stereotaxic frame (Kopf Instruments, Tujunga, California, United States of America) on a heating pad to prevent hypothermia. We used eye protection gel against drying. The scalp in the midline was removed (from Bregma to Lamda) and the skull thoroughly cleaned and treated with 3% hydrogen peroxide for implantation.

## Viral injection

A craniotomy was drilled above the PVT and the virus was slowly injected (1 nl/min) into the PVT (AP: −0.9, ML: 0, AP and ML taken from the bregma). We injected 50 to 50 nl in 2 dorso-ventral positions (DV: 3.1 and 2.9, DV taken from the brain surface) in the behavioural, corti-costerone, c-Fos, and GABA-A receptor localization experiments and 150–150 nl in the same positions in case of the tetrode experiments. Virus was allowed to express for a minimum of 3 weeks before behavioural and tetrode experiments (S2A, S3A, S6, S9, S11 and S12 Figs).

## Implantation

In the behavioural, corticosterone, c-Fos, and EEG experiments, 2 optic fibres were implanted above the PVT (AP: −0.9 and −1.4, ML: 0, DV: 2.8, 10˚ angle), together with 3 EEG screw elec-trodes (2 above the frontal lobe, at AP: +0.2, ML: +/− 1.8 and 1 above the parietal lobe at AP: −1.1, ML: +/− 1.8) and 1 EMG wire electrode in the neck muscle (directly behind the skull). The EEG and EMG electrodes were soldered to Omnetics connectors (Omnetics). In the tet-rode experiments, tetrodes were also implanted above the PVT (AP: −0.9, ML: 0, DV: 2.7, 9.5˚-angle) and gradually screwed down to PVT until the start of the experiment. Optical fibres, EEG and EMG electrodes were fixed with dental cement (Metabond), tetrode wires were fixed with flexible glue first and then also with cement (Figs 2A, 3A, 4A, 5A, 6A, 8A, S6 and S9). After surgery, the animals were given saline and painkillers for 1 to 5 days according to their condition. Recovery from surgery lasted 2 weeks.

## Recording apparatus

All mice were recorded in single-animal boxes with sound and electrical noise isolation. Cam-eras (FlyCapture) and LEDs were built in the boxes, providing an ambient 200 lux (50 cm from the cage) light. Electrophysiological recordings were provided via Intan Interface Board, Intan cables and chips (Intan Technologies). To avoid entanglement of wires and fibres optics as well as to reduce the weight of the cables and connectors, a dual (wire and fibre optic) com-mutator system was used (Imetronic), connecting to the Intan-based recording system and the head implants of the animals. We recorded the electrophysiological signal with a custom-made script of the Bonsai visual-programming software.

## Predatory odour stress exposure (POSE)

We assessed defensive behaviour to an ecologically pertinent aversive stimulus namely expo-sure to predator odour stress (POSE), using a synthetic analogue of a fox anogenital product (2-methyl-2-thiazoline; 2MT). The assessments were conducted in a transparent plexiglass arena measuring 32 × 12.5 × 15 cm. The testing environment featured a fume hood with back-ground sound set at 62 dB and high-intensity light ranging from 280 to 320 lux. To standardise odour exposure across subjects, testing occurred in covered cages. At start, mice were placed in the testing cage and allowed to freely explore for a 2-min baseline period. This initial period served as a reference point. Subsequently, each subject was exposed to fox odour (2-MT) for a duration of 10 min [39,40]. For this presentation, 2 µl of pure, undiluted 2-MT (sc-251779, Santa Cruz Biotechnology) was pipetted onto a 2 × 2 cm filter paper, which was then placed into a 1.5 ml Eppendorf tube. After a 2-min baseline period, the Eppendorf was placed into the left corner of the test cage. Mouse behaviour was recorded (Bonsai) and quantified (Solomon) using a video-based measurement system. We used 2 behavioural variables to characterise the innate behavioural responses: freezing and escape jumps. The mice were considered to freeze if movement was not detected for 2 s. Identical procedure was used for the NOE experiments,

but in this case 2-MT was omitted. In case of home cage controls, the animals were left in their home cages until the beginning of the perfusion.

## Behavioural protocol

In the behaviour experiments, we distinguished 3 major states: the pre-stress period (5 days PRE 1–5 days), the day of POSE (POST 0 day), and the post-stress period (4 days POST 1–4 days) (Figs 1A, 2A, 3A, 4A, 5A, 6A, 8A, S6 and S9). In case of the LATE group, we made the optogenetic inhibition 5 days after the POSE and then we continued recording in the next 5 consecutive days, the same way as detailed above (Fig 8A).

Animals were habituated to the recording apparatus for 3 days (3 h/day) prior to the experiments. The home cages of the animals were placed to sound insulated boxes, their lids were removed, and the cables were connected to the head implants of the animals. Animals were allowed to move freely and to climb to the edge of their home cages. During the pre-stress periods, the baseline activity of the mice (EEG/EMG and behaviour in $n = 28$ animals, PVT/CR+ unit activity and behaviour in $n = 6$ animals) were recorded for 3 h every day starting at the beginning of their light phase (ZG time 0) for 5 days. On the day of the POSE animals were transferred to a new room and were placed into a new cage for POSE (see above). After POSE, we put the animals back to their come cages and continued recording for the next 3 h. In the first hour, the laser protocol was performed, which meant inhibition in case of the SwiChR group. On the POST period, we recorded the activity of the mice in the same way as on the PRE period. POST period lasted for 4 days.

## Tetrode recording

In the tetrode experiments, movable drives were built and implanted, and 28 channels of unit activity, 1 channel of EEG, and 1 channel of EMG were recorded (Figs 2A, 4A and 6A).

The tetrode drives consisted of 7 tetrodes (28 channels), an optical fibre (105 μm core diameter, 0.22 NA, Thorlabs, New Jersey, USA), an EMG wire electrode, EEG screw electrode, a ground, and a reference cable attached to the electrode interface board. The tetrodes were positioned 300 micrometres below the tip of the optical fibre. The end of the optical fibre was milled into a cone shape. The tetrodes were gold-plated the day before implantation to a resistance below 80 k ohms

In tetrode recording, animals were subjected to the same behavioural test as described above (see in Behavioural protocol section). Phototagging was used to identify PVT/CR+ cells (see in Optogenetic inhibition and phototagging section). Data were recorded with OpenEphys board and software. Video was recorded in the same way as in the behavioural experiments (see in Behavioural protocol section).

## Optogenetic inhibition and phototagging

In all experiments, immediately after the POSE the animals were returned to their home cage and 2 external optical fibres (105 μm core diameter, 0.22 NA, Thorlabs, New Jersey, USA) were coupled to the implanted optical fibres with a ceramic sleeve. A rotary joint (Doric Lenses, Québec, Canada) was built in the commutator system to avoid the coiling of optical fibres and Intan cables. Optical manipulations were provided with a 473 nm DPSS laser (LaserGlow Technologies). Before every use laser intensity was checked by a manual photometer (Thorlabs, New Jersey, USA). Laser intensity was also monitored online during the stimuli via built in photosensors (Thorlabs, New Jersey, USA). Lasers were controlled via Matlab with a National Instrument Board (Matlab) and Arduino (Arduino Uno). Photoinhibition was maintained by laser pulses and the same laser protocol was performed in case of the SwiChR group

(stressed and inhibited) and also in EYFP group (stressed) for 60 min. Both for EYFP (stressed, $n = 43$ mice) in SwiChR (stressed and inhibited $n = 30$ mice) mice trains of 2 s ON and 13 s OFF light pulses were used (10 to 15 mW).

For phototagging in ChR2-expressing mice ($n = 4$), we utilised 4 Hz, 1 ms long pulses (0.5 to 5 mW; 120 to 150 stimulation) at the beginning and at the end of the recordings (S4D Fig). For phototagging in SwiChR animals ($n = 2$), we used 4, 2 s long (2 s ON and 13 s OFF) pulses (0.5 to 5 mW) during wakefulness as well as during sleep (4–4 trains) (S1 Fig).

## Histological verification of viral infection and optic fibre/tetrode placement

Mice were anesthetised with 2% isoflurane followed by an intraperitoneal injection of an anesthetic mixture (ketamine, 83 mg/kg; xylazine, 3.3 mg/kg) to achieve deep anesthesia. Next, mice were perfused transcardially with 0.9% saline for 2 min followed by 100 ml of 4% PFA solution. After perfusion, brains were removed from the skull, and thalamic section was immersion-fixed in 2% PFA for 30 min. Brains were cut into 50 μm sections using a vibrating microtome (Vibratome 3000). To verify the viral infection and cannula/fibre optic placement every third sections were stained from each animal. Sections were washed 4 times in 0.1 M phosphate buffer (PB) and incubated in a blocking solution containing 2% TritonX-100 and 10% normal donkey serum (Jackson ImmunoResearch, USA) in PB for 2 h at room temperature. Sections were then incubated (24 h) with anti-calretinin guinea pig (1:1,000, 214104, Synaptic Systems, RRID: AB_10635160), anti-GPF chicken (1:1,000, A10262, Thermo Fisher Scientific, RRID: AB_2534023), and anti-IBA1 rabbit (1:500, 27030, Wako, RRID: AB_2314667) primary antibodies dissolved in 2% normal donkey serum solution. Afterward, sections were washed in PBS 4 times (10 min each time), then incubated with secondary antibodies. The primary antibodies were visualised using the following secondary antibodies: Alexa Fluor 488-conjugated donkey anti-chicken (1:1,000, 703-545-155, Jackson ImmunoResearch), Cy3-conjugated donkey anti-rabbit (1:1,000, 711-165-152, Jackson ImmunoResearch), and Alexa Fluor 647- conjugated donkey anti-guinea pig (1:1,000, 706-605-148, Jackson ImmunoResearch) antibodies. After being washed with PBS (phosphate buffer saline), the sections were mounted on glass slides and cover slipped using Mowiol 4–88 (81381, Sigma-Aldrich) fluorescent mounting medium. Images were acquired on Panoramic Digital Slide Scanner (Zeiss, Plan-Apochromat 10X/NA 0.45, xy: 0.65 μm/pixel, Panoramic MIDI II; 3DHISTECH, Budapest, Hungary).

## c-Fos mapping with corticosterone measurement

Following POSE, neuronal activation and stress hormone reactivity were investigated using c-Fos immunohistochemistry and serum corticosterone measurement, respectively (Figs 1A and S2). The following experimental groups were included:

(1) Home cage control group (Home Cage, $n = 7$). These animals were left in their home cage for the whole duration of the experiment. They were not moved to a new cage on the day of the experiment, received no POSE or laser stimulation.

(2) No odour exposure (NOE) group ($n = 5$). These animals were moved to the new cage, were not exposed to 2 MT there but received laser stimulation when they returned to their home cage using the same protocol as the other 2 groups.

(3) EYFP group ($n = 7$). These animals were moved to the new cage, were exposed to 2 MT, and received laser stimulation when they returned to their home cage. Since these animals were injected with EYFP containing virus PVT/CR+ cell did not experience optogenetic inhibition.

(4) SwiChR group (*n* = 6). These animals were moved to the new cage, were exposed to 2 MT, and received laser stimulation when they returned to their home cage. The activity of PVT/CR+ cells were reduced by optogenetic inhibition (Fig 7A).

All groups were submitted the same habituation protocol to familiarise the mice with the recording environment (for details, see Behavioural protocol section). Mice were decapitated 60 min after POSE. Trunk blood was collected for CORT radioimmunoassay (RIA) and brains were fixed in 4% PFA for 3 days. c-Fos, CR, GFP triple staining was performed on 50-μm thick coronal sections selected by a 1:6 fractionator sampling covering the whole rostral to caudal extension of PVT and the forebrain regions targeted by PVT/CR+ neurons including prelimbic cortex (PrL, +2.22 mm to +1.54 mm from Bregma), nucleus accumbens shell (NAcS, +1.94 mm to+ 0.86 mm from Bregma), paraventricular hypothalamic nucleus (PVH, –0.58 mm to –1.22 mm from Bregma), central amygdala (CeA, –1.22 mm to –1.94 mm from Bregma), basolateral amygdala (BLA, –0.58 mm to –1.94 mm from Bregma), and bed nucleus of the stria terminalis (BNST, 0.26 mm to 0.02 mm from Bregma).

For immunocytochemistry, free-floating brain sections were washed in 0.1 M PB and blocked in 10% normal donkey serum (NDS) in PB containing and 2% Triton-X 100 for 2 h at room temperature. The primary antibodies against c-Fos (rabbit, 1:1,000, 226003, Synaptic Systems, RRID: AB_2231974), calretinin (guinea pig, 1:1,000, 214104, Synaptic Systems, RRID: AB_10635160), and GFP (chicken, 1:1,000, A10262, Thermo Fisher Scientific, RRID: AB_2534023) were diluted in PB containing 2% NDS. Twenty-four hours later, sections were washed multiple times and were incubated with Alexa Fluor 488-conjugated donkey anti-chicken (1:1,000, 703-545-155, Jackson ImmunoResearch), Cy3-conjugated donkey anti-rabbit (1:1,000, 711-165-152, Jackson ImmunoResearch), Alexa Fluor 647-conjugated donkey anti-guinea pig (1:1,000, 706-605-148, Jackson ImmunoResearch) antibodies, and with Hoechst 33258 (1:2,000, Sigma-Aldrich) for 2 h at room temperature. After further PB washes, sections were mounted on glass slides and coverslipped using Mowiol 4–88 (81381, Sigma-Aldrich) fluorescent mounting medium. Images were taken using a Panoramic Digital Slide Scanner (Zeiss, Plan-Apochromat 10X/NA 0.45, xy: 0.65 μm/pixel, Panoramic MIDI II; 3DHISTECH, Budapest, Hungary). Every third section was stained and analysed per animal. To assess c-Fos activation in the PVT/CR+ neuron projections targets, we delineated and identified regions of interest (PrL, NAcS, BLA, and CeA) based on axonal labelling and on neuroanatomical landmarks using Hoechst counterstaining on fluorescent pictures applying CaseViewer 2.3 software (3DHISTECH). c-Fos signal was counted bilaterally in covering the whole anteroposterior extension of the actual region/subregion.

To assess the POSE-induced activation of PVT/CR+ neurons, c-Fos immunoreactivity of CR expressing cells were analysed manually in 20× confocal images below the optic fibre using a Nikon A1R confocal microscope (CFI Plan Apo VC20X/N.A. 0.75, xy: 0.31 μm/pixel, Nikon Europe, Amsterdam, the Netherlands).

To quantify c-Fos density throughout the anteroposterior extension of the PVT, anti-c-Fos (rabbit; 1:20,000, PC38, Calbiochem, RRID: AB_2106755) was developed with DABNi as a chromogen. All sections used for quantification were developed together for the same duration. The section was dehydrated and then mounted with DePex (Serva, Heidelberg, Germany). The number of c-Fos-labelled cells was analysed using a custom-written ImageJ (RRID:SCR_003070) script.

## Corticosterone measurement

Serum corticosterone was measured by direct RIA as described [74]. Briefly, the corticosterone antibody was raised in rabbits against corticosterone-carboxymethyloxime bovine serum

albumin conjugate. [125]I-labelled corticosterone-carboxymethyloxime-tyrosine methyl ester was used as tracer. The interference from plasma transcortin was eliminated by inactivating transcortin at a low pH. Assay sensitivity was 1 pmol, the intraassay coefficient of variation was 7.5%.

## GABA-A receptor subunit expression analysis

To investigate the effect of stress on GABAA receptor subunit expression levels in the PVT, we used SwiChR ($n = 4$) and EYFP ($n = 3$) injected animals which underwent POSE as described previously (Figs 7A and S11); 24 h after POSE, injected and not injected home cage control (Home Cage, $n = 3$) mice were anesthetised with 2% isoflurane followed by an intraperitoneal injection of an anesthetic mixture (ketamine, 83 mg/kg; xylazine, 3.3 mg/kg) to achieve deep anesthesia. Next, mice were perfused transcardially with 0.9% saline for 2 min followed by 100 ml of 1% PFA solution. After perfusion, brains were removed from the skull, and thalamic section was immersion-fixed in 2% PFA for 30 min. Brains were cut into 50 μm sections using a vibrating microtome (Vibratome 3000). Perfusion-fixed sections were washed 4 times in 0.1 M PB for 1 h. For detection of GABA$_A$-receptor γ2 and α1 subunits, sections were pretreated with 0.2 M HCl (hydrogen-cloride) solution containing 0.2 mg/ml pepsin (Dako) at 37˚C for 2 min, followed by a 2 h long blocking period in 10% normal donkey serum in PB as described previously [75]. This was followed by an overnight incubation at room temperature in a solution containing 2% NDS, 0.05% sodium azide, and a mixture of primary antibodies: rabbit anti-γ2 (1:2,000, 224003, Synaptic System, RRID:AB_2263066) [75], guinea-pig anti-α1 (1:500, MSFR101550, GABAARa1-GP, Frontier Institute) [76], and chicken anti-GFP (1:1,000, A10262, Thermo Fisher Scientific, RRID:AB_2534023) to enhance the endogenously expressed EYFP signal. The primary antibodies were visualised using the following secondary antibodies: Alexa Fluor 488-conjugated donkey anti-chicken (1:1,000, 703-545-155, Jackson ImmunoResearch), Cy3-conjugated donkey anti-rabbit (1:1,000, 711-165-152, Jackson ImmunoResearch), and Alexa Fluor 647-conjugated donkey anti-guinea pig (1:1,000, 706-605-148, Jackson ImmunoResearch) antibodies. After being washed with PBS, the sections were mounted on glass slides and cover slipped using Mowiol 4–88 (81381, Sigma-Aldrich) fluorescent mounting medium. Images from PVT were obtained using a Nikon A1R confocal microscope (Plan Apo VC 60× oil objective; numerical aperture, 1.40; xy, 0.10 μm/pixel; z-step size, 0.10 μm). Images were taken through a z-plane (1.5 μm) at the centre of the tissue containing 20 stacks within each region. During image acquisition gain, offset, laser intensity, zoom, and pinhole were kept constant. In each animal, 12 to 30 regions of interest (ROIs, rectangular frame) restricted to the PVT were analysed from 3 brain sections. To determine the expression levels of GABA$_A$-receptor subunits, the images were thresholded and gray values of pixels were registered in ROIs defined in appropriate optical sections of image stacks in ImageJ software (RRID:SCR_003070).

## Ex vivo slice preparations

Ex vivo electrophysiology was performed on adult CR/IRES-Cre mice ($n = 4$) infused in the PVT with the AAV5.EF1a.DIO.SwiChRCA.TS.EYFP.WPRE virus (Fig 2A). Coronal brain slices (300 μm) including the PVT were prepared minimum 4 weeks following surgery. Following anesthesia with isoflurane, the brain was quickly removed in an ice-cold bicarbonate-buffered solution (BBS) containing the following (mM): 116 NaCl, 2.5 KCl, 1.25 NaH$_2$PO$_4$, 26 NaHCO$_3$, 30 glucose, 1.6 CaCl$_2$, 1.5MgCl$_2$, and $5 \times 10{-}5$ minocycline (pH 7.4, bubbled with 95% O$_2$, 5% CO$_2$). Slices were then cut using a vibrating blade microtome (Campden model 7000smz2) in an ice-cold solution containing (mM): 150 sucrose, 10 choline-Cl, 2.5 KCl, 3.1

Na-pyruvate, 11.6 ascorbic acid, 1.25 $NaH_2PO_4$, 26 $NaHCO_3$, 30 glucose, 0.5 $CaCl_2$, 7 $MgCl_2$, supplemented with D-APV (25 μm). They were then transiently immersed (10/15 min) in a bath containing the same solution at 32˚ to 34˚C (pH 7.4, bubbled with 95% $O_2$, 5% $CO_2$). Finally, slices were transferred to warm BBS (32˚ to 34˚C) for the rest of the experimental day.

In our hands, SwiChR was tonically activated by ambient light. Throughout the experimental day, we thus paid particular attention to protect slices from all light sources.

For electrophysiology, slices were moved to a recording chamber mounted on an upright Olympus BX51WI microscope (Olympus France) and continuously perfused with bubbled BBS (~1/1.5 ml/min; 32 to 35˚C). Cells were visualised with a combination of Dodt contrast and an on-line video contrast enhancement. Whole-cell (WC) recordings of PVT/CR+ cells were performed with an EPC-10 double amplifier (Heka Elektronik, Lambrecht/Pfalz, Germany) run by the PatchMaster software (Heka). PVT cells could be easily identified in the transmitted deep red light (∼750 nm) with which slices were visualised using a CoolSnap HQ2 CCD camera (Photometrics, Trenton, New Jersey, USA) run by MetaMorph (Universal Imaging, Downington, Pennsylvania, USA).

Whole-cell (WC) recordings were performed with pipettes (resistance: 2/3 MΩ) filled with an intracellular solution containing (mM): 75 NMDG, 4.6 $MgCl_2$, 10 HEPES, 0.5 EGTA, 4 $Na_2$-ATP, 0.4 $Na_2$-GTP, 1 QX-314, ~300 mOsm, and pH 7.3 with HCl. This solution maximises the driving force for chloride ions by containing a close to symmetrical concentration with respect to the external BBS.

Spontaneous inhibitory postsynaptic currents (IPSCs) were recorded at a holding potential of −70 mV in the continuous presence of the AMPA receptor blocker NBQX (5/10 μm) in the bath. Only SwiChR-expressing neurons were recorded, identified by an inward current developing in response to 470 nm diode pulses (Thorlabs Maisons-Laffitte, France). Light pulses were relayed to the recording chamber via the epifluorescence pathway of the microscope. During recordings, the series resistance was partially compensated (max 65%), whereas liquid junction potentials were not corrected. The analysis was performed using custom-built routines in Igor (Wavemetrics, Lake Oswego, USA).

## Data analysis

**EEG/EMG analysis.**   EEG and EMG data processing was carried out using Matlab. EEG signals were down-sampled at 2 kHz and low-pass filtered for 50 Hz for further analysis. Power of delta (1 to 4 Hz) frequency band were calculated from one of the frontal screw electrodes. EMG signal was detected either directly from the EMG electrode or indirectly from one of the parietal EEG screw electrodes. For further analysis, EMG signal was downsampled at 2 kHz and bandpass filtered between 300 and 600 Hz. Sleep-wake states were determined using EEG and EMG signal. Wake was characterised by high EMG activity, and low delta power, while NREM sleep was characterised by low muscle tone and high delta power.

EMG ON and OFF states were defined in the following way. EMG time series were divided to 0.1 sec bins and standard deviation (SD) was calculated for each bin. Plotting a probability distribution for SD values of muscle activity for each animal, we were able to determine a value (peak of the distribution) characteristic for muscle activity in sleep. Then, using a threshold— determined for each animal (+/− [2.1–2.5] SD of baseline)—each time bin was assigned either EMG ON (1 value) or EMG OFF (0 value) [13].

## Sleep electroencephalogram (EEG) spectral slope analysis

Periods of NREM sleep EEG spectral slope was defined according to a previously published approach [50]: the best linear fit to the equidistantly oversampled (piecewise cubic Hermite

interpolation) double logarithmic plot of the average power spectral density (Welch periodogram based on 4 s Han windowing and Fast Fourier Transformation). Fitting was based on the 2 to 6 Hz and 18 to 48 Hz ranges, in order to exclude frequencies characterised by frequent upward deflections caused by NREM sleep-specific spectral peaks. The slope of the linear is known to reflect the depth of sleep (sleep intensity) [50,77].

### Horizontal activity

The horizontal movement of the animals was calculated from the video. Using a neural network-based program package called DeepLabCut, we calculated the X and Y coordinates of the animal's head for each video frame (Figs 3B, lower panel; 3C, lower panel; 5B, lower panel; and 5C, lower panel). DeepLabCut's neural network was pre-trained on our own data. We then filtered the coordinates based on their likelihood (at least 0.8) and calculated the magnitude of displacement for each frame. For Figs 3D, 5D and 8D, we averaged 15, 1-min long displacement data. For Figs 3E, 5E and 8E, the displacement data set was normalised by the averaged nest onset time of each animal calculated for the PRE period.

### Analysis of behaviour in the nest

We defined the following behavioural variables:

Nest onset: Nest onset was defined as the moment when an animal enters its nest and did not leave it for at least 20 s.

Sleep onset: We defined sleep onset as the time point when the first 2 min of uninterrupted sleep was initiated. Sleep onset was defined using the combination of EEG and EMG (3 times larger amplitude of the EEG activity compared to baseline and EMG OFF state).

Nest time: Nest time was defined as the time between nest onset and sleep onset.

For the detailed analysis of the behaviour, we used the Solomon Coder software from the beginning of the recording until the sleep onset of the animals. During the analysis, we differentiated and quantified the following behaviours:

Nest building: A stereotyped mouse behaviour, during which the mouse usually reaches out, pulls in or gathers nest material to build its nest (S5 Movie).

Freezing in the nest: During this freezing-like behaviour in the nest, the animals were awake; the body of the animal did not perform horizontal movements for at least 2 s (range 2 to 60 s). During this type of behaviour, the animals sometimes perform sudden jerky movements, head-bobbing or turning but no nest building, tramping, nest tromping, comfort movements or rearing (S6 Movie) [78,79].

Hyperventilation: On POST 0 day, immediately after the stress animals displayed hyperventilation visible in the video [80]. This was detected as a rhythmic movement of the thorax interspersed with short whole body movements. This type of activity could be detected only on POST 0 day (S3 Movie).

### Unit clustering

The clustering was done in a semi-automatic way. The 3 h of measurement were recorded in 2 files of 1.5 h each. These files were automatically analysed with SpyKING Circus. SpyKING Circus automatically generated clusters. These clusters were then manually curated using the Phy program [81]. All clusters with an average amplitude below 50 μV were discarded (S4A and S4E Fig). Noise was then filtered out from the clusters by displaying the different PCA component features of the channels and cutting the resulting clusters (S4E Fig). In doing so, it was checked that a given unit really belonged to a single cell and that the same cell was not scattered in several clusters (using Amplitude View and Feature View). Then, the

autocorrelograms (S4C Fig) and inter spike interval distributions of the cells were checked (no spike within 2 ms and ISI distribution with Gaussian curve shape were accepted).

## Unit activity analysis

**State modulation index (SMI).**   All recordings were split into 3 behavioural stages based on the camera recordings, namely "Wake," "Nest," and "Sleep." For sleep, we get an approximately identical time window to Wake and Nest state (first 20 to 30 min of sleep), this way in unit data sleep consists mainly NREM stages. The average firing rate of each neuron was computed for each stage, normalised to its time and to the average firing rate of each state on the PRE period. To quantify the effect of behavioural state on the firing of the PVT/CR+ neurons, we defined a SMI as: (a-b)/(a+b) where "a" is the proper firing rate of the PVT/CR+ neuron in state with higher activity of the comparison (i.e. wake or nest) and "b" is the proper firing rate of the cell with lower activity (i.e., nest or sleep).

**High-frequency activity (HFA).**   HFA was defined as a series of 2 or more action potentials with interspike intervals less than 10 ms (S7A Fig). Autocorrelograms were calculated at 1 ms resolution for each neuron. Normalised values were defined as the ratio of spikes in high-frequency clusters within all spikes (S7C Fig).

**Correlated activity (CA).**   CA was calculated for all simultaneously recorded pairs of neurons. Crosscorrelograms were calculated for each pair at 1 ms resolution. Correlation events between cell pairs were defined as the spikes of a neuron within ±5 ms with a lag from the spikes of the other neuron (S7B Fig). To characterise how likely is that pairs of neurons fire together in a ±5 ms time window, we calculated the mean counts in the ±5 ms lags of the CCGs and normalised these values with the mean counts in the 180 to 200 ms baseline period (S7D Fig).

**Cell tracking among days.**   For identification of the same cell on a given tetrode between days, we took the average action potential shape of each cell for each channel of the tetrode. To do this, we clipped the signal shape around the cell action potential peak times in a ±2 ms window, sampled at 15 kHz from the original signal high-pass filtered at 250 Hz. Then, we identified the channel with the highest peak amplitude and used this value to normalise the waveforms with the largest peak amplitude in each 4 channels of the tetrode. We quantified the waveform similarity between all recorded single unit activities by taking the sum of the squared time point-by-time point differences in the 4 channels (S3B Fig). Finally, we examined the distribution of these points and identified the most similar 5% of the pairs as neurons recorded more than once on consecutive days. Similarity score values ranged from 0.0148 $\mu V^2$ to 80.3909 $\mu V^2$. As a cutoff value, we took the lowest 5% of the similarity scores which was 2.75 $\mu V^2$ (S3C Fig). In total, we analysed 1,404 unit pairs across the 4 animals, of which 334 pairs met our similarity score criteria. For this manuscript, we included exclusively those units that were recorded before the stress (PRE 5 days) and immediately after the stress (POST 0 days).

**Optotagging with ChR2.**   Stimulus-associated spike latency test (SALT) was used to identify significantly light-activated PVT/CR+ neurons in control animals (ChR2) [82]. This test determines whether light pulses significantly changed the spike timing of a neuron by comparing the distribution of first spike latencies relative to the light pulse, assessed in a 10-msec window after light-stimulation, to 10-msec epochs in the baseline period (−150 to 0 msec from the onset of light-stimulation). Neurons activated at $p < 0.01$ were considered identified CR+ neurons.

Optotagging with SwiChR: During SwiChR tagging, we used the protocol, detailed above in Optogenetic inhibition and phototagging section. For each cell, we calculated the firing rate in

the second before laser ON and in the 2nd second of the laser ON period and computed the change between them using the formula (a-b)/(a+b) was used. We measured inhibition using this ratio and considered as inhibited if this value was above 0.1 (S1 Fig).

## Statistics

Data are represented as mean ± SEM. All the statistical analysis was done using GraphPad Prism version 9. First, normality of the data distribution was checked, using Shapiro–Wilk normality test. When comparing 2 groups, if both groups showed a normal distribution, a *t* test was used. Paired *t* tests were used when the same animal or cell was tested over several days (indicated by a connecting line between the 2 points in the figures) and unpaired *t* tests in all other cases. If the requirements of *t* test were not fulfilled, then Mann–Whitney test or Wilcoxon matched-pairs signed rank tests were used. To compare multiple groups, one-way ANOVA was used followed by Tukey's post hoc test. The significance level was set at $p < 0.05$. Individual data points are shown in the figures. All the statistics used in figures and supplementary figures are shown in tables of statistics (S10 Data).

## Supporting information

**S1 Fig. Effect of SwiChR photoinhibition on the activity of PVT/CR+ neurons during tetrode recordings.** (A) Representative example of the firing rate of a PVT/CR+ neuron during 2 s laser ON 13 s laser OFF trains ($n = 9$ trains) in the wake state. Top Raster plot of one neuron during 9 trains. Blue rectangle marks the laser ON time (2 s). Bottom Firing rate for the same cell calculated in 10 ms windows. (B) Z-scored firing rate of 8 phototagged PVT/CR+ neurons using the $4 \times 15$ s phototagging protocol (see Methods). Mean (blue line) +/- SD (grey) are shown. Straight blue line above marks the laser ON time (2 s). (C) Z-scored firing rate of 9 phototagged PVT/CR+ neurons during the 60 min long photoinhibition protocol (see Methods). Mean (blue line) +/- SD (grey) are shown. Straight blue line above marks the laser ON time (2 s).
(TIFF)

**S2 Fig. Histology of the behavioural experiments, c-Fos expression across the rostro-caudal extent of the PVT and EEG sleep slopes on POST 0 days.** (A) Schematics of coronal sections illustrating the location of the optical fibres (yellow dots) and the extent of transfection following SwiChR (top, blue) and EYFP (bottom, green) virus constructs injected to the PVT of CR-Cre mice. Drawings are based on a compilation of 14 animals for EYFP and 7 for SwiChR. The schematics of coronal sections were created according to the Franklin and Paxinos mouse brain atlas [83]. Right, representative images depicting fibre optic tracks from SwiChR and EYFP injected CR-Cre mice. Green represent GFP immunolabelling, magenta represents CR immunolabelling. (B) Representative images showing c-Fos immunolabelling in the PVT with fibre optic tracks (blue arrows) from home cage control, EYFP, and SwiChR mice after POSE. Created with BioRender.com. (C) Quantification of c-Fos expression across the rostro-caudal extent of the PVT. Blue arrows mark the position of the fibre optics (−0.88 Bregma level, $F_{(2,10)} = 58.33$, $p = 0.0001$; −1.15 Bregma level, $F_{(2,13)} = 12.44$, $p = 0.001$; −1.42 Bregma level, $F_{(2,11)} = 18.53$, $p = 0.0003$). (D) Comparison and the steepness of the NREM EEG spectral slope between the fifth day (black dots) of the pre-stress period and POST0 day (red dots) in EYFP mice ($t_{[13]} = 3.58$, $p = 0.0034$). Dots represent individual animals. Underlying data can be found in S9 Data. See S10 Data for the full results of the statistical tests. Data are means ± SEM. $^*p < 0.05$, $^{**}p < 0.01$, $^{***}p < 0.001$, $^{****}p < 0.0001$.
(TIFF)

**S3 Fig. Histological verifications of the tetrode experiments and tracked neurons from PRE5 day to POST0 day in ChR2 animals.** (A) Schematic of coronal sections illustrating the placement of tetrodes (yellow lines) and extent of injection sites using conditional ChR2 or SwiChR (blue) containing virus constructs targeted to the PVT in CR-Cre mice. Drawings are based on 6 mice ($n = 4$ for ChR2 and $n = 2$ for SwiChR). The schematic of coronal sections was created according to the Franklin and Paxinos mouse brain atlas [83]. (B) Representative waveforms of optotagged PVT/CR+ neurons in different tetrodes of the same animal shown in Fig 2E (tetrodes: TT2, TT7, TT8). The neurons were recorded for 2 consecutive days (top vs. bottom row). Channel numbers (Ch) and similarity scores (see Methods) between the 2 days are shown next to the waveforms. (C) Distribution of similarity scores of all tagged PVT/CR+ neurons from the ChR2 animals. Red line indicates the threshold. Left, all cell pairs. Right, zoomed in image showing the distribution of cell pair numbers within the range of the threshold.
(TIFF)

**S4 Fig. Representative example of the clustering process in case of 3 individual neurons recorded on the same tetrode in the PVT.** (A) Action potentials (arrows, coloured by yellow, orange, and purple) of 3 PVT units identified on a 50-ms long raw trace (high pass filtered, above 150 Hz). (B) Timing of action potentials of the same 3 neurons on a 1-s long window (two of them is CR+, one of them are putative CR-). (C) Autocorrelograms of these 3 PVT neurons, calculated from the 3 h long recording with 1 ms bins in a 50 ms window. (D) Peristimulus time histogram of the optogenetic stimulation (tagging) of the same 3 neurons during wakefulness in the 30 ms before and after the stimulus. Red ticks mark the onset of the 1 ms long stimulation. One non-tagged (yellow, putative CR-) and 2 tagged (orange and purple, CR+) neurons are shown. (E) The 3 amplitude clusters of the same 3 neurons in the 3 channels used for clustering (one of the PCA components during clustering).
(TIFF)

**S5 Fig. Temporal dynamics of predator odour stress-induced behavioural alterations.** (A) Day by day normalised horizontal locomotor activity of EYFP animals during the PRE 1–5 and the POST 1–4 days (one-way RM-ANOVA followed by Fisher post hoc test, $F(8,101) = 2.065$, $p < 0.05$; PRE1 vs. POST4 $p < 0.05$; PRE2 vs. POST4 $p < 0.05$; PRE3 vs. POST4 $p < 0.05$; PRE4 vs. POST1 $p < 0.05$; PRE4 vs. POST3 $p < 0.01$; PRE5 vs. POST2 $p < 0.05$; PRE5 vs. POST4 $p < 0.01$). (B) Awake nest time duration of EYFP animals during the PRE 1–5 and the POST 1–4 days (one-way RM-ANOVA followed by Fisher post hoc test, $F(8,98) = 2.542$, $p < 0.05$; PRE1 vs. POST4 $p < 0.05$; PRE2 vs. POST4 $p < 0.05$; PRE3 vs. POST4 $p < 0.05$; PRE4 vs. POST3 $p < 0.01$; PRE4 vs. POST4 $p < 0.01$; PRE5 vs. POST3 $p < 0.01$; PRE5 vs. POST4 $p < 0.01$; POST1 vs. POST3 $p < 0.05$; POST1 vs. POST4 $p < 0.05$). (C) Time spent with nest building by EYFP animals during the PRE 1–5 and the POST 1–4 days (one-way RM-ANOVA $F(8,99) = 1.272$, $p > 0.05$). (D) Time spent with freezing-like behaviour by EYFP animals during the PRE 1–5 and the POST 1–4 days (one-way RM-ANOVA followed by Fisher post hoc test, $F(8,100) = 5.774$, $p < 0.0001$; PRE1 vs. POST4 $p < 0.001$; PRE2 vs. POST1 $p < 0.01$; PRE2 vs. POST3 $p < 0.01$; PRE2 vs. POST4 $p < 0.0001$; PRE3 vs. POST1 $p < 0.01$; PRE3 vs. POST3 $p < 0.01$; PRE3 vs. POST4 $p < 0.0001$; PRE4 vs. POST1 $p < 0.001$; PRE4 vs. POST2 $p < 0.05$; PRE4 vs. POST3 $p < 0.01$; PRE4 vs. POST4 $p < 0.0001$; PRE5 vs. POST1 $p < 0.05$; PRE5 vs. POST4 $p < 0.001$; POST2 vs. POST4 $p < 0.01$).
(TIFF)

**S6 Fig. Long-term behavioural consequences of exposure to a novel environment (NOE) without predator odour.** (A) Scheme of the experiment. EYFP-injected CR-Cre mice were

exposed to a novel environment without predator odour to assess the effects of environmental novelty on home cage behaviour before (PRE) and after novelty exposure (POST). Created with BioRender.com. (B) Schematics of coronal section depicting the extent of transfection following EYFP (green) virus constructs injected to the PVT of CR-Cre mice. (C) The averaged, normalised horizontal locomotor activity of EYFP-injected CR-Cre mice exposed before nest onset (t[4] = 1.642, $p$ = 0.1759). Dots represent the averaged daily values of individual animals. (D) Temporal dynamics of averaged, normalised horizontal locomotor activity of EYFP the animals ($n$ = 5) during the PRE1–5 days (black) vs. the POST1–4 days (grey) periods. (E–G) (E) Nest time (t[4] = 0.7887, $p$ = 0.4744), (F) nest building time (t[4] = 0.8094, $p$ = 0.4637), (G) freezing time (t[4] = 1.965, $p$ = 0.1208) in EYFP mice ($n$ = 5) during the PRE and POST period. Dots represent the averaged daily values of individual animals. Underlying data can be found in S9 Data. See S10 Data for the full results of the statistical tests. Data are shown as mean ± SEM.
(TIFF)

**S7 Fig. Representative examples and normalised values of high-frequency activity (HFA) and correlated activity (CA) in ChR2 tagged neurons during the PRE and POST periods.** (A) Autocollerograms of 3 tracked neurons with 50 ms lag recorded on the PRE 5 day (before the stress) and on the POST1 day (after the stress) during wake state. The red shading shows the time window (−10; 10 ms) used to compute HFA values (shown above the autocorrelograms). Note pronounced change in HFA activity (large central peaks) in Neuron 2 after the stress. (B) Crosscorrelograms of 3 tracked pairs of neurons with 200 ms lag on the PRE 5 day (before the stress) and on the POST1 day (after the stress). The red shading shows the time window within (−5; 5 ms) used to compute CA values (shown below the crosscorrelograms). CA values increase in all 3 pairs after the stress. (C) HFA values normalised with mean firing rate (see Methods) of the recorded and tagged neurons from control (ChR2) animals on the PRE ($n$ = 69 from 4 animals) and the POST ($n$ = 57 neurons from 4 animals) period (wake, U = 1,912, $p$ = 0.792; nest, U = 1,786, $p$ = 0.3789; sleep U = 1,910, $p$ = 0.7845). (D) CA values normalised with baseline activity (see Methods) of the recorded and tagged neurons from control (ChR2) animals on the PRE ($n$ = 69 from 4 animals) and the POST ($n$ = 57 neurons from 4 animals) period (U = 15,337, $p$ = 0.8632; nest, U = 12,969, $p$ = 0.0082; sleep U = 14,073, $p$ = 0.1357). Underlying data can be found in S9 Data. See S10 Data for the full results of the statistical tests. Data are means ± SEM. ***$p$ < 0.01.
(TIFF)

**S8 Fig. Temporal dynamics of stress-induced behavioural consequences following post stress photoinhibition of PVT/CR+ neurons.** (A) Day by day normalised horizontal locomotor activity of SwiChR animals during the PRE 1–5 and the POST 1–4 days (one-way RM-ANOVA followed by Fisher post hoc test, F(8,44) = 1.129, $p$ = 0.363). (B) Awake nest time duration of SwiChR animals during the PRE 1–5 and the POST 1–4 days (one-way RM-ANOVA followed by Fisher post hoc test, F(8,45) = 1.161, $p$ = 0.344). (C) Time spent with nest building by SwiChR animals during the PRE 1–5 and the POST 1–4 days (one-way RM-ANOVA F(8,45) = 0.816, $p$ = 0.596). (D) Time spent with freezing-like behaviour by SwiChR animals during the PRE 1–5 and the POST 1–4 days (one-way RM-ANOVA followed by Fisher post hoc test, F(8,45) = 0.6272, $p$ = 0.7505).
(TIFF)

**S9 Fig. Long-term behavioural consequences of PVT/CR+ inhibition without predator odour exposure.** (A) Scheme of the experiment. SwiChR-injected CR-Cre mice were exposed to a novel environment without predator odour followed by 1 h long photoinhibition in the

home cage. Behavioural variables were measure in the home cage behaviour before (PRE) and after novelty exposure (POST). Created with BioRender.com. (B) Schematics of coronal section depicting the extent of transfection following SwiChR (green) virus constructs injected to the PVT of CR-Cre mice. IBA1 immunostaining highlights the placement of the optic fibres. (C) The averaged, normalised horizontal locomotor activity of SwiChR-injected CR-Cre mice exposed before nest onset (t[4] = 1.539, $p$ = 0.198). Dots represent the averaged daily values of individual animals. (D) Temporal dynamics of averaged, normalised horizontal locomotor activity of SwiChR the animals ($n$ = 5) during the PRE1–5 days (black) vs. the POST1–4 days (blue) periods. (E–G) (E) Nest time (t[4] = 2.429, $p$ = 0.0721), (F) nest building time (t[4] = 3.253, $p$ = 0.0313), (G) freezing time (t[4] = 3.327, $p$ = 0.0292) in SwiChR mice ($n$ = 5) during the PRE (black) and POST (blue) period. Dots represent the averaged daily values of individual animals. Underlying data can be found in S9 Data. See S10 Data for the full results of the statistical tests. Data are shown as mean ± SEM. *$p$ < 0.05.
(TIFF)

**S10 Fig. Normalised values of high-frequency activity (HFA) and correlated activity (CA) in SwiChR tagged neurons during the PRE and POST periods.** (A) Normalised HFA values with firing rate (see Methods) of the recorded and tagged neurons from inhibited (SwiChR) animals on the PRE ($n$ = 38 from 2 animals) and the POST ($n$ = 39 neurons from 2 animals) period (wake, U = 870.5, $p$ = 0.5301; nest, U = 891, $p$ = 0.6496; sleep U = 862, $p$ = 0.4854). (B) Normalised CA values with baseline activity (see Methods) of the recorded and tagged neurons from control (SwiChR) animals on the PRE ($n$ = 38 from 2 animals) and the POST ($n$ = 39 neurons from 2 animals) period (U = 7,306, $p$ = 0.552; nest, U = 7,418, $p$ = 0.6884; sleep U = 7,356, $p$ = 0.6106). Underlying data can be found in S9 Data. See S10 Data for the full results of the statistical tests. Data are means ± SEM.
(TIFF)

**S11 Fig. Injection sites and fibre optic locations of mice involved in GABA A receptor and c-Fos mapping experiments.** Schematics of coronal sections illustrating the location of the optic fibres (yellow dots) and the extent of transfection following SwiChR (top, blue) and EYFP (bottom, green) virus constructs injected to the PVT of CR-Cre mice. Drawings are based on a compilation of 10 animals for EYFP and 10 for SwiChR. The schematics of coronal sections were created according to the Franklin and Paxinos mouse brain atlas.
(TIFF)

**S12 Fig. Injection sites and fibre optic locations of mice involved in LATE photoinhibition experiment.** Schematics of coronal sections illustrating the location of the optical fibres (yellow dots) and the extent of transfection following SwiChR virus constructs injected to the PVT of CR-Cre mice. Drawings are based on a compilation of $n$ = 7 SwiChR mice. The schematics of coronal sections were created according to the Franklin and Paxinos mouse brain atlas.
(TIFF)

**S13 Fig. Temporal dynamics of stress-induced behavioural alterations: effects of photoinhibition on PVT/CR+ Neurons applied five days post-POSE.** SwiChR injected mice underwent home cage recordings preceding and following POSE. Photoinhibition was applied after 5 days following POSE. (A) Day by day horizontal locomotor activity of SwiChR expressing LATE mice during the POST 1–4 and the LID 1–5 days (one-way RM-ANOVA followed by Fisher post hoc test, F(8,47) = 3.453, $p$ = 0.0033; POST1 vs. LID1, $p$ < 0.05; POST2 vs. LID1, $p$ < 0.001; POST2 vs. LID2, $p$ < 0.01; POST2 vs. LID3, $p$ < 0.001; POST2 vs. LID4, $p$ < 0.01; POST2 vs. LID5, $p$ < 0.01; POST4 vs. LID1, $p$ < 0.05; POST4 vs. LID2, $p$ < 0.05; POST4 vs. LID3, $p$ < 0.05. (B) Awake nest time duration of SwiChR expressing LATE mice during the

POST 1–4 and the LID 1–5 days (one-way RM-ANOVA, $F_{(8,48)} = 1.475$, $p > 0.05$). (C) Time spent with nest building by SwiChR expressing LATE mice during the POST 1–4 and the LID 1–5 days (one-way RM-ANOVA $F_{(8,48)} = 1.59$, $p > 0.05$). (D) Time spent with freezing-like behaviour by SwiChR expressing LATE mice during the POST 1–4 and the LID 1–5 days (one-way RM-ANOVA followed by Fisher post hoc test, $F_{(8,48)} = 3.686$, $p < 0.01$; POST1 vs. LID2, $p < 0.05$; POST1 vs. LID3, $p < 0.01$; POST1 vs. LID4, $p < 0.05$; POST1 vs. LID5, $p < 0.05$; POST2 vs. POST4, $p < 0.05$; POST3 vs. LID3, $p < 0.05$; POST4 vs. LID1, $p < 0.01$; POST4 vs. LID2, $p < 0.001$; POST4 vs. LID3, $p < 0.001$; POST4 vs. LID4, $p < 0.01$; POST4 vs. LID5, $p < 0.001$).
(TIFF)

**S1 Movie. Two representative animals from the EYFP group during the predator odour stress exposure (POSE) protocol displaying strong defensive behaviour, mostly freezing and escape jumps.** The capsules containing 2MT are located in lower part of the cage. POSE is performed in a cage distinct from the home cage in a different room, under a fume hood.
(MP4)

**S2 Movie. Two representative animals from the no odour exposure (NOE) group in the same environment.** Animals display normal explorative behaviour.
(MP4)

**S3 Movie. Behaviour of an animal from the EYFP group displaying hyperventilation and freezing, immediately after POSE when the animal was returned to its home cage.**
(MP4)

**S4 Movie. Behaviour of an animal from the SwiChR group displaying normal locomotor behaviour, immediately after POSE when the animal was returned to its home cage.** The 2 s light pulse on the head implant indicates the laser ON periods.
(MP4)

**S5 Movie. Characteristic nest building behaviour of an animal from the EYFP group during the PRE period.** The animal collects, processes, and arranges the paper used as nesting material.
(MP4)

**S6 Movie. Characteristic behaviour of an animal from the EYFP group during the POST period.** The animal mainly displays freezing-like behaviour (shortly freezing) in the nest, signalled by a red square at the bottom of the movie. During this behaviour, the animal remains largely stationary, refrains from horizontal movements (for at least 2 s) and may exhibit sudden jerky movements, head-bobbing, or turning. Nest building or rearing are rare.
(MP4)

**S1 Data. Supporting data for Fig 1.**
(XLS)

**S2 Data. Supporting data for Fig 2.**
(XLS)

**S3 Data. Supporting data for Fig 3.**
(XLS)

**S4 Data. Supporting data for Fig 4.**
(XLS)

**S5 Data. Supporting data for Fig 5.**
(XLS)

**S6 Data. Supporting data for Fig 6.**
(XLS)

**S7 Data. Supporting data for Fig 7.**
(XLS)

**S8 Data. Supporting data for Fig 8.**
(XLS)

**S9 Data. Supporting data for S2, S6, S7, S9 and S10 Figs.**
(XLS)

**S10 Data. Supporting data for the full results of the statistical tests.**
(XLS)

## Acknowledgments

We thank the Light Microscopy Center and the Virus Technology Unit at the Institute of Experimental Medicine for kindly providing microscopy support. Authors would like to thank Krisztina Faddi, Dániel Kuti, Kornél Demeter, and Csaba Dávid for their excellent technical assistance. The schemes were created with www.biorender.com.

## Author Contributions

**Conceptualization:** Anna Jász, László Acsády.

**Data curation:** Anna Jász, László Biró, Bálint Király, Orsolya Szalárdy, László Acsády.

**Formal analysis:** Anna Jász, László Biró, Zsolt Buday, Bálint Király, Orsolya Szalárdy.

**Funding acquisition:** Róbert Bódizs, Krisztina J. Kovács, Marco A. Diana, Balázs Hangya, László Acsády.

**Investigation:** Anna Jász, László Biró, Zsolt Buday, Bálint Király, Krisztina Horváth, Gergely Komlósi, Krisztina J. Kovács, Marco A. Diana, László Acsády.

**Methodology:** Anna Jász, László Biró, Zsolt Buday, Bálint Király, Orsolya Szalárdy, Krisztina Horváth, Róbert Bódizs, Balázs Hangya.

**Project administration:** Anna Jász, László Biró, László Acsády.

**Resources:** Anna Jász, Bálint Király, Balázs Hangya.

**Software:** Anna Jász, Bálint Király, Balázs Hangya.

**Supervision:** László Acsády.

**Validation:** Anna Jász, László Acsády.

**Visualization:** Anna Jász, László Biró, Bálint Király.

**Writing – original draft:** Anna Jász, László Biró, László Acsády.

**Writing – review & editing:** Anna Jász, László Biró, Gergely Komlósi, Róbert Bódizs, Krisztina J. Kovács, Marco A. Diana, Balázs Hangya, László Acsády.

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
