## [Editor Report · Decision Letter 0]

22 Mar 2024

Dear Dr Biro, 

Thank you for submitting your manuscript entitled " Persistently increased post-stress activity of paraventricular thalamic neurons is essential for the emergence of stress-induced alterations in behavior" for consideration as a Research Article by PLOS Biology.

Your manuscript has now been evaluated by the PLOS Biology editorial staff as well as by an academic editor with relevant expertise and I am writing to let you know that we would like to send your submission out for external peer review.

Once your full submission is complete, your paper will undergo a series of checks in preparation for peer review. After your manuscript has passed the checks it will be sent out for review. To provide the metadata for your submission, please Login to Editorial Manager (https://www.editorialmanager.com/pbiology) within two working days, i.e. by Mar 24 2024 11:59PM.

Kind regards,

Christian

Christian Schnell, PhD

Senior Editor

PLOS Biology

cschnell@plos.org

---

## [Decision Letter · Decision Letter 1]

24 May 2024

Dear Dr Biro,

Thank you for your patience while your manuscript " Persistently increased post-stress activity of paraventricular thalamic neurons is essential for the emergence of stress-induced alterations in behavior " was peer-reviewed at PLOS Biology. It has now been evaluated by the PLOS Biology editors, an Academic Editor with relevant expertise, and by several independent reviewers. 

In light of the reviews, which you will find at the end of this email, we would like to invite you to revise the work to thoroughly address the reviewers' reports.

As you will see below, the reviewers think that the study is well executed and provides important insights. However, they also raise a couple of concerns regarding the lack of methodological details, missing control experiments, and incomplete discussion of the existing literature. 

In addition to that, we received advice from a reviewer who was not able to submit a full report due to time constraints but we still think that these points would be important to address. I have included these comments, which partially overlap with the concerns from the other reviewers, as comments Reviewer 4 below. 

Given the extent of revision needed, we cannot make a decision about publication until we have seen the revised manuscript and your response to the reviewers' comments. Your revised manuscript is likely to be sent for further evaluation by all or a subset of the reviewers.

**IMPORTANT - SUBMITTING YOUR REVISION**

*Re-submission Checklist*

*Published Peer Review*

*PLOS Data Policy*

*Blot and Gel Data Policy*

Sincerely,

Christian

Christian Schnell, PhD

Senior Editor

PLOS Biology

cschnell@plos.org

REVIEWS:

Reviewer #1: This study investigates the initial steps leading to enduring changes in behavior caused by acute stress exposure. The authors show that a single episode stressor increased the firing activity of calretinin-positive neurons within the PVT, which persisted for several days. When the authors inhibited these neurons for 1 hour after the stress exposure, it attenuated the increase in firing rate as well as the stress-induced behavioral changes. This manipulation also reduced activation of downstream targets associated with defensive behaviors, BLA, prelimbic and nucleus accumbens. Surprisingly, photoinhibiting the same neurons 5 days after the stress produced similar attenuations. The data are convincing that PVT/CR+ neurons are persistently changed after stress leading to altered behaviors. I do have some questions, mostly related to the analyses and interpretations.

1. In Fig 2e, the authors show 3 sample neurons across 2 sessions, pre and post stress, with a calculated "similarity score" for each neuron. How was this score calculated? What does a score 0.3 really mean? In the methods (LINE 1080), the authors report the similarity score ranged from "0 and 1.32". Presumably 0 means not similar. Why were these neurons included in the pre/post stress analyses? Should there not be a cutoff criterion for a neuron's similarity score to be included in the analyses? In Suppl 3c, I am not understanding the y-axis of just "histogram" and the x-axis says "similarity score" but shows a range of 0 to 100, when actual similarity scores were 0-1.32. What does the red threshold marker mean? Threshold to calculate a similarity score? Threshold to be included in the analyses? 

2. For phototagging, what was the criterion for a phototagged cell? Within how many ms did the neuron need to respond to be considered phototagged? Is it possible that CR- neurons that received input from CR+ cells were included as phototagged CR+ cells? The x-axis in Suppl 4d is also not labeled.

3. For the experiment where photoinhibition was applied 5 days after the stress leading to reversal of many of the behaviors, etc is it not surprising that a single photoinhibition session so many days after the stressor could have such long-lasting effects? The first photoinhibition experiment, where it was applied right after the POSE, makes more sense because it is presumably interfering with the consolidation/induction of the hyperactivity. However, once the hyperactivity was established how does only 1 photoinhibition session seemingly reverse everything? There is next to nothing about this in the discussion. 

4. In Fig 8b,c, what is the data normalized to? Fig 8c shows LID has reduced horizontal activity and yet Fig 8e shows reduced freezing compared to POST. These don't seem to make sense together. How can there be reduced movement and reduced freezing? What is being analyzed differently between these 2 metrics?

5. The authors mention in the introduction that the PVT/CR+ neurons have diurnal alterations in activity. Since the authors had long term recordings, including wake/sleep data, did the authors observe disruptions to the diurnal rhythm of these neurons' activity? Given the other data presented, it seems that this would happen. Related, did the sleep stages change after stress (i.e. non-REM vs REM).

6. A prior study (Hua et al 2018, Current Biology) also looked at PVT/CR+ neurons in sleep/wake disturbances after starvation (a type of chronic stress) and showed the PVT-BNST pathway to be important. Can the authors comment on why BNST was not included in the present study?

7. The authors do a nice job describing CR+ neurons of the PVT and their distribution and function. It might also be worth adding to the intro and/or discussion an overview of calretinin itself and its function and how it is able to support persistently higher neural activity. 

8. LINE 382-3: Something is weird about the grammar of this particular sentence. 

9. Figs 2, 3, and 4 have a schematic showing laser photoinhibition for 1h after POSE, but it does not appear that any of the data in these figures are from photoinhibited animals?

10. I could not find "defensive behaviors" defined for Fig 1d.

Reviewer #2: 

This paper explores the role of the paraventricular nucleus of the thalamus (PVT) in the long-term effects of traumatic events, employing a combination of predator odor stimulation and long-term neural activity recordings via tetrodes. While the role of the PVT in stressed behaviors has been extensively studied, previous research has primarily focused on event-locked activities, with few analyzing longitudinal activity. Although I cannot confirm if this paper is indeed the first to report long-term neural activation, the authors present a unique approach and successfully elucidate PVT activities in enduring stressed behaviors.

Major points:

A) Virus reporter specificity: The authors should address concerns regarding the specificity of the virus reporters, particularly Calretinin/Calb2-Cre, which are not exclusively expressed in the PVT but also in neighboring thalamic nuclei. Supplementary Figures 3, 10, and 11 indicate significant non-PVT expression of Cre reporters. The authors should carefully analyze the "off-target" expression and its implications, including the positions of infected somas and the expression of c-fos and GABA-R in non-PVT CR+ nuclei. Discussion on the potential effects of off-target expressions is warranted, as strong off-target effects could shift the study's focus to the midline thalamus rather than the PVT.

B) Regional Functionality: While previous reports highlight functional and characteristic distinctions between anterior and posterior PVT, the current manuscript lacks mention of regional specificity and utilizes manipulation between aPVT and pPVT. The authors should discuss regional specificity of PVT and justify their approach. Elaborating on Supplemental Figure 2C could strengthen their claim by revealing which part of PVT (and other CR+ nuclei) is activated by POSE and inhibited by SwitChR.

C) Lack of controls: Several control experiments are necessary to support the assertion that post-stress event inhibition of PVT cells is critical. These include 1) assessing the effects of photoinhibition without POSE to show short/long-term effect of PVT inhibition itself ,2) determining the timing of inhibition by inhibition in pre- and during POSE, and 3) conducting long-term observations with YFP-infected animals (for Fig 8) to assess natural recovery from POSE stress.

Minor points:

a) Inconsistent figure labeling: There is inconsistency between figure panel labeling and the main text, with small case letters used in figure panels and large case letters in the legend.

b) Figure 1: There are two panel Ls, and panel M is missing.

c) Clarity on experimental timeline: From Figure 3 onwards, it is unclear what "Pre" and "Post" represent in each experiment. Specifying whether these refer to averages or results from specific days would enhance clarity.

d) "CR-" neurons in Supplementary Figure 4: Should these be referred to as "ChR2-"? Clarification on distinguishing CR+ neurons without AAV infection and CR- neurons is needed.

e) Line 559: Clarify whether "Same as H" refers to "same as B."

Reviewer #3: Dr Jasz and colleagues present an interesting set of experiments focused on the role of the PVT in translating transient, acute stress to sustained behavioral llterations. Building on their prior work, they show that a 10 minute exposure to novelty coupled with a predator odour changes both the firing rate of calretinin cells in the PVT and stress/defensive behaviors. The show altered activity of the CR/PVT cells, and an effect of silencing these cells post hoc on subsequent behaviors. They suggest that the CR/PVT neuronal ensembles engage in protracted increased firing which, in turn, influences the consequences of stress.

The paper present an interesting concept an a significant amount of work. However, several points deserve the authors' attention.

First, the analyses are complex ad confusing. The authors are requested to put side by side the pre- and poststress behaviors after eYFP or inhibitory CR/PVT stimulation, and provide a two way ANOVA to look explicitly at the effects of stress and of the optogenetic manipulation. Without this, one is left trying to piece out facts in figures 1 and 5, which do not enable direct comparisons. Not clear why the authos use ttests and one way ANOVA. 

A second general question pertains to the relation of the current work with the Matyas 2018 paper. There, CR neurons in the dorsomedial thalamus are implicated in arousal. Interestingly, some demonstrate prolonged firing in response to a tail pinch. The term PVT is not used in the 2018 paper, though clearly, the CR/PVT neurons are involved. Might the authors clarify the relation of the two papers, and the novelty of the current work? 

Note that the authors move mice from the home cage to a new one when presenting a predator odour. Thus, they generate arousal together with stress. In view of the role of anterior PVT in arousal, this is a confounder. While the authors show that the new cage by itself does not generate stress behaviors, they cannot claim that the novelty is unimportant. Indeed, they are measuring the combined effects of predator odour and novelty. 

They choose a single type of arousal/ stress, is this generalizable to others?

Both Bhatnagar and Do Monte suggest posterior PVT is involved in stress encoding. Where in PVT did the authors implant their hardware? They provide coordinates, but in view of the striking divergence of different atlases for delineating the PVT, can the authors specify what part of that elongated structure they tackled? 

While the authors identify an interesting potential mechanism for a more enduring effect of PVT on stress, they seem to be unaware of recent work using genetic tagging, which shows that PVT encodes transient stress early-in life, and translates it to enduring changes in adult behavior. This work should be acknowledged.

In view of the complexity and heterogeneity of the PVT, the authors are requested to provide actual photomicrographs of their targeting, rather than schematics

While the authors ascribe important functions for the augmented activity in PVT CR cells do they know what instigates it? What stops it? A speculation in the discussion would be helpful. The changes in GABAA alpha 1 and gamma 2 by themselves do not answer these questions

Reviewer 4:

The topic of this manuscript seems interesting. But, I found a few potential major issues:

* The stress paradigm: The “stress” is exposure to a predator odor. There are several articles (e.g., Staples, 2008; McGregor, 2002; Hacquemand, 2010) suggesting that fox odor (in contrast to cat saliva or other cat odors) is not really a fear-inducing stressor but just a bad smelling odor, especially at high concentration (youused 100%). The 2-Methyl-2-thiazoline, which is used instead of the commonly-studied fox odor chemical TMT (2,4,5-Trimethylthiazole), has been shown to produce strong freezing responses (Isosaka, 2015 PMID: 26590419). Is this odor, that is not a component of fox urine, acting as a bad smell rather than an innate fear signal?

Figure 1 controls: You compare post-stress behavior (hyperventilation and sleep onset) in EYFP to SwiChR mice, but do not have unstressed mice to compare to. Comparing pre to post stress is problematic for many of these analyses, because animals also change over time and with repeated handling and testing. Please compared to unstressed controls for many of these analyses.

Figure 5 compared to Figure 3: You test SwiChR mice pre and post stress in Figure 5, but do not compare these results directly to the EYFP control animals shown in Figure 3. A change in one group and lack of change in another does not demonstrate a difference between groups. Please directly compare these groups statistically.

---

## [Decision Letter · Decision Letter 2]

15 Nov 2024

Dear László,

Thank you for your patience while we considered your revised manuscript "Persistently increased post-stress activity of paraventricular thalamic neurons is essential for the emergence of stress-induced alterations in behavior" for publication as a Research Article at PLOS Biology. This revised version of your manuscript has been evaluated by the PLOS Biology editors, the Academic Editor and the original reviewers.

Based on the reviews and on our Academic Editor's assessment of your revision, we are likely to accept this manuscript for publication, provided you satisfactorily address the following data and other policy-related requests. After discussing Reviewer 3's comment with the Academic Editor, we don't think you need to address Reviewer 3's remaining comment. However, please address the following points:

* Please add the links to the funding agencies in the Financial Disclosure statement in the manuscript details.

* DATA POLICY:

Regardless of the method selected, please ensure that you provide the individual numerical values that underlie the summary data displayed in the following figure panels as they are essential for readers to assess your analysis and to reproduce it: 1DEFGHIKLM, 2CDFGHIJK, 3DEFGHIJ, 4CEGI, 5DEFGHIJ, 6BCDE, 7CDEFGIJ, 8BCDEFG, S2CD, S6CDEFG, S7CD, S9CDEFG and S10AB.

* CODE POLICY

* Please note that per journal policy, the model system/species studied should be clearly stated in the abstract of your manuscript. 

We expect to receive your revised manuscript within two weeks. 

*Published Peer Review History*

*Press*

Sincerely,

Christian

Christian Schnell, PhD

Senior Editor

cschnell@plos.org

PLOS Biology

Reviewer remarks:

Reviewer #1: All concerns have been adequately addressed.

Reviewer #2: The authors adequately responded to all of the points I raised. and I do not have any further comments on the updated manuscript. 

Reviewer #3: The authors largely responded comprehensively to my concerns.

I do recommend including the 2-WAY ANOVA analyses in the paper.

---

## [Editor Report · Decision Letter 3]

2 Dec 2024

Dear László,

Thank you for the submission of your revised Research Article "Persistently increased post-stress activity of paraventricular thalamic neurons is essential for the emergence of stress-induced alterations in behavior" for publication in PLOS Biology. Thank you also for clarifying my questions regarding Figure 6B and the revised source data file.

On behalf of my colleagues and the Academic Editor, Heather Cameron, I am pleased to say that we can in principle accept your manuscript for publication, provided you address any remaining formatting and reporting issues. These will be detailed in an email you should receive within 2-3 business days from our colleagues in the journal operations team; no action is required from you until then. Please note that we will not be able to formally accept your manuscript and schedule it for publication until you have completed any requested changes.

PRESS

Sincerely, 

Christian

Christian Schnell, PhD

Senior Editor

PLOS Biology

cschnell@plos.org